# CRISPR/Cas9-mediated targeted mutagenesis of *GmTCP19L* increasing susceptibility to *Phytophthora sojae* in soybean

**Sujie Fan**[1,2☯], **Zhuo Zhang**[1☯], **Yang Song**[1], **Jun Zhang**[1], **Piwu Wang**[1]*

**1** Plant Biotechnology Center, College of Agronomy, Jilin Agriculture University, Changchun, Jilin, People's Republic of China, **2** Crop Science Post-doctoral Station, Jilin Agricultural University, Changchun, Jilin, People's Republic of China

☯ These authors contributed equally to this work.
* peiwuw@163.com

**Data Availability Statement:** All relevant data are within the paper and its Supporting Information files.

## Abstract

The TEOSINTE BRANCHED1/CYCLOIDEA/PROLIFERATING CELL FACTOR (TCP) transcription factors is one of the superfamilies of plant-specific transcription factors involved in plant growth, development, and biotic and abiotic stress. However, there is no report on the research of the TCP transcription factors in soybean response to *Phytophthora sojae*. In this study, *Agrobacterium*-mediated transformation was used to introduce the CRISPR/Cas9 expression vector into soybean cultivar "Williams 82" and generated targeted mutants of *GmTCP19L* gene, which was previously related to involve in soybean responses to *P. sojae*. We obtained the *tcp19l* mutants with 2-bp deletion at *GmTCP19L* coding region, and the frameshift mutations produced premature translation termination codons and truncated GmTCP19L proteins, increasing susceptibility to *P. sojae* in the T2-generation. These results suggest that *GmTCP19L* encodes a TCP transcription factor that affects plant defense in soybean. The new soybean germplasm with homozygous *tcp19l* mutations but the BAR and Cas9 sequences were undetectable using strip and PCR methods, respectively, suggesting directions for the breeding or genetic engineering of disease-resistant soybean plants.

## Introduction

TEOSINTE BRANCHED1/CYCLOIDEA/PROLIFERATING CELL FACTOR (TCP) transcription factors (TFs) are plant-specific transcription factors family. There were named based on the first four identified members: TEOSINTE BRANCHED1 (TB1) from maize (*Zea mays*), CYCLOIDEA (CYC) from snapdragon (*Antirrhinum majus*), and PROLIFERATING CELL FACTORS 1 and 2 (PCF1 and PCF2) from rice (*Oryza sativa*) [1, 2]. The members of TCP transcription factors family all contain conserved domains and encode structurally similar proteins [3–5]. The conserved TCP domain, a noncanonical basic-Helix-Loop-Helix (bHLH) domain, composed of 59 amino acids, is shown to play vital roles in DNA binding and the interaction between proteins [3]. According to the sequence differences in the TCP domains, the 24 TCPs in *Arabidopsis thaliana* can be classified into Class I (PCF or TCP-P)

**Funding:** SF was supported by National Natural Science Foundation of China (31801381, http://www.nsfc.gov.cn/) and Education Department of Jilin Province (JJKH20180658KJ, http://jyt.jl.gov.cn/), ZZ was supported by Education Department of Jilin Province (JJKH20190935KJ, http://jyt.jl.gov.cn/), YS was supported by Jilin Scientific and Technological Development Program (20190103120JH, http://kjt.jl.gov.cn/), The funders had no role in study design, data collection and analysis, decision to publish, or preparation of the manuscript.

**Competing interests:** The authors have declared that no competing interests exist.

and Class II (TCP-C) subfamilies. Class I members promote cell proliferation and growth, but class II members inhibit these processes [6, 7]. The class II members can be further divided into two subcategories, CINCINNATA (CIN)-like TCP (CIN-TCP) and CYCLOIDEA/TEOSINTE BRANCHED1 (CYC/TB1) [5]. TCP transcription factors modulate target gene expression by specifically binding to the *cis*-acting element, and the consensus binding sequences for class I is GGNCCCAC while class II is G(T/C)GGNCCC [4, 6].

TCP transcription factors are widely present in plants and play important roles in growth development, morphogenesis, hormone synthesis, signal transduction, and the response to biotic and abiotic stresses [3, 8–19]. In Arabidopsis, a large number of TCP transcription factors have been found, and most of which are related to the formation of leaves, flowers and axillary buds, and the synthesis of plant hormones [12, 20–24]. AtTCP3 can interact with MYB, bHLH and WD40 (MBW) transcription factors to regulate the synthesis of anthocyanins [25]. AtTCP14 and AtTCP15 can modulate cell proliferation in the developing leaf blade and specific floral tissues [26], and also participate in the induction of genes involved in gibberellin biosynthesis and cell expansion by high temperature functionally [27, 28]. AtTCP12/AtTCP18 can integrate into the FLOWERING LOCUS T (FT)–FD complex to control floral initiation and also directly bind the promoter of downstream floral meristem identity gene *APETALA1*(*AP1*) to enhance its transcription and regulate flowering [7]. Experimental evidence indicates that several TCP transcription factors can be rapidly induced in response to altered environmental conditions and to integrate hormonal signals [4, 29–31]. Based on transcriptome data analysis and related experiments, it is found that TCP transcription factors are likely to regulate the differential expression of auxin-induced genes [4, 31, 32].

In addition, recent studies supported that TCP transcription factors can also play an important role in plant defense signaling networks by stimulating the synthesis of certain active metabolites, such as brassinosteroid, jasmonic acid (JA) and flavonoids [20, 21, 25, 33, 34]. More importantly, in the process of plant innate immune response, TCP transcription factors were found to be involved in the effector-triggered immunity (ETI) of pathogenic bacteria [22, 35, 36]. In Arabidopsis, AtTCP13, AtTCP15 and AtTCP19 can be targeted by effectors from the gram-negative bacteria *Pseudomonas syringae* and the oomycete *Hyaloperonospora arabidopsidis*, and the plants mutated in *AtTCP13*, *AtTCP15* or *AtTCP19* exhibit altered infection phenotypes [35]. AtTCP8, AtTCP14, AtTCP15, AtTCP20, AtTCP22 and AtTCP23 can interact with suppressor of rps4-RLD1 (SRFR1, a negative regulator of ETI) and regulate the expression of defense-related genes which are hostile to SRFR1, facilitating plant disease resistance [22]. The secreted AY-WB protein 11(SAP11), an effector protein factor of *Xanthomonas oryzae*, can specifically target CIN-like TCP transcription factor, inhibiting the expression of *LOX2* gene and reducing the production of JA, and improving the reproduction ability of insects [33, 37].

In a previous study, a *TCP* gene (Glyma.05G050400.1) was significantly induced after infection with *P. sojae* among several soybean near isogenic lines revealed by comparative transcriptomics [38]. In this study, the full-length of this *TCP* gene was isolated using RT-PCR technique. It showed the highest homolgy with AtTCP19 protein after comparison with the 24 TCP transcription factor members of Arabidopsis, and this *TCP* gene was named as *GmTCP19-Like* (*GmTCP19L*). Subsequently, we employed the CRISPR/Cas9 system to specifically induce targeted mutagenesis of the *GmTCP19L* in the soybean cultivar "Williams 82" and studied the function of this gene responses to *P. sojae* infection. We obtained a variety of homozygous *tcp19l* mutants with short deletions in T1 generation, and the targeted mutations were stably inherited from the T1 to T2 generation. In T2-generation, the homozygous mutants for null alleles of *GmTCP19L* frameshift mutated by a 2-bp deletion and the BAR and Cas9 sequences were undetectable using strip and PCR methods, respectively, which resulted in enhanced susceptibility to *P. sojae* infection by decreasing the activity of antioxidant defense

system. Our findings suggest that *GmTCP19L* directly or indirectly regulates soybean resistance to *P. sojae*. These mutants of *GmTCP19L* that we obtained will provide materials for more in-depth research on *GmTCP19L* functions and the molecular mechanism responses to *P. sojae* infection in soybean.

## Materials and methods

### Plant material, growth condition and strain material

"Williams 82", a *P. sojae*-resistant soybean cultivar carrying resistance gene *Rps1k*, was used for transformation in this study. Seeds collected from the T0 generation were sown in pots filled with sterile vermiculite in a growth chamber with a 14-h photoperiod (at a light intensity of 350 mol m$^{-2}$s$^{-1}$) at 22°C/18°C day/night temperatures and relative humidity of 70 ± 10%.

  *P. sojae* race 1, PSR01, which is the dominant race in Jilin Province, was kindly provided by Professor Shuzhen Zhang and her team (Soybean Research Institute, Key Laboratory of Soybean Biology of Chinese Education Ministry, Northeast Agricultural University, Harbin, China). It was isolated from infected soybean plants in Heilongjiang [39].

### sgRNA design and CRISPR/Cas9 expression vector construction

The soybean endogenous gene *GmTCP19L* (Glyma.05G050400.1) sequence was downloaded from the Phytozome database (http://www.phytozome.net/). Potential sgRNA target sites within the *GmTCP19L* gene were identified using the web tool CRISPR-P (http://cbi.hzau.edu.cn/crispr/) [40]. The primer binding sites for amplification of selected sgRNAs target sites were designed using Primer Premier 5.0. Functional domain was predicted using CDSearch [41]. To construct the *GmTCP19L*-CRISPR/Cas9 vector carrying both *GmTCP19L* targeted sgRNA and Cas9 cassettes, the sequence of Cas9 was assembled downstream of the CaMV 35S promoter together with the sgRNA driven by the *Glycine max* U6 promoter (GmU6) within its T-DNA region, the *bar* gene driven by a CaMV 35S promoter was used as a screening marker (**S1** and **S2** **Figs**).

### Plasmid delivery by tri-parental mating and soybean transformation

The *GmTCP19L*-CRISPR/Cas9 plasmids were transformed into *Agrobacterium tumefaciens* strain EHA105 via tri-parental mating. *A. tumefaciens* EHA105 was grown on LB agar with Rif selection for 36–48 h at 28°C. *E. coli* DH5α harboring *GmTCP19L*-CRISPR/Cas9 plasmids and *E. coli* HB101 harboring pRK2013 plasmids were grown on LB agar witn Kan selection for 12 h at 37°C. The bacteria were scraped off their respective plates and resuspended in at least 1 mL of LB medium each with no antibiotics. Combine 100 μL of each pre-culture into a reaction tube and pellet the cells in a microcentrifuge at 1,000×g for 1 min at room temperature. Resuspend the cells in 20–30 μL of LB medium and placed them at the center of a LB agar plate with no antibiotic selection at 28°C for 3–5 h. Use an inoculation loop to take the biomass up from the agar surface and resuspend them in 1 mL of LB medium containing no antibiotics. 100 μL of this suspension was then streaked out on a LB agar plate supplemented with Rif and Kan at 28°C for 72 h until clear colonies have formed. The cotyledonary nodes from "Williams 82" were used as explants for tissue culture and soybean transformation using the *Agrobacterium-mediated* transformation method described by Guo et al. [42].

### Screening for mutations by sequencing analysis

Genomic DNA was extracted from the leaves of each individual plant following the modified cetyltrimethylammonium ammonium bromide (CTAB) protocol in the T0, T1 and T2

generation, and then the 621 bp *GmTCP19L* target region was amplified via PCR with Phanta_Super Fidelity DNA Polymerase (Vazyme Biotech). PCR products were detected by 1% agarose gel electrophoresis and then sequenced using the *GmTCP19L*-F and *GmTCP19L*-R primers. Three types of gene editing can be identified by sequence peaks. The heterozygous mutations showed overlapping peaks at the target site, and the wild type (WT) and homozygous mutations showed single peaks at the target. The homozygous mutations were identified by sequence alignment with the WT sequence, and short base insertions or deletions induced by CRISPR/Cas9 can lead to frameshift mutations. At the same time, we also screened the *tcp19l* mutants without the BAR and Cas9 sequences of the CRISPR/Cas9 vector in both T1 and T2 progenies. PAT/Bar test strip (Catalog No. STX 14200, Agdia, USA) was used to identify the BAR protein following the manufacturer's instruction, and the primers Cas9-F/R were used to amplify the fragment (349 bp) of the Cas9 (S2 Fig).

## Analysis of off-target mutagenesis

To examine if either of target sites could have off-target activity, we analysed the potential off-target sites using online website tool CRISPR-P (http://cbi.hzau.edu.cn/crispr/). Two most potential off-target sites at *GmTCP19L*-SP1 and *GmTCP19L*-SP2 were selected, and the sequences were downloaded from the Ensembl Plants database (http://plants.ensembl.org/Glycine_max/). Primers of each off-target site were designed to amplify 300–500 bp regions. The regions spanning the target sites were amplified by PCR technique, then the different types of potential off-target sites editing can be identified by sequencing analysis.

## Resistance identification of the *tcp19l* mutants

To examine the phenotype of *tcp19l* mutants in response to *P. sojae* infection, the T1 living cotyledons at the first-node stage (V1) were treated with *P. sojae* as described by Ward *et al.* (1979) with minor modifications for the live plants. To determine whether the *tcp19l* mutants can transmit the phenotype to their progenies, the progeny of *tcp19l* mutants without BAR and Cas9 sequences were sown in pots filled with nutritious soil and grown in a greenhouse at Jilin Agriculture University under previously described conditions. *P. sojae* race 1 was cultivated on V8 juice agar in a petri dish at 25˚C for 7 days. For disease resistance analysis, hypocotyls of fourteen-day old seedlings were inoculated using the wounded-hypocotyl inoculation technique [43, 44]. For the same purpose, roots of fourteen-day old seedlings were inoculated with zoospores using a hydroponic assay according to the method of Ward *et al.* (1979) and Gijzen *et al.* (1996) with minor modifications [45, 46]. The inoculated seedlings were grown in a mist chamber at 25˚C with 90% relative humidity under a 14-h photoperiod at a light intensity of 350 mol m$^{-2}$ s$^{-1}$ [43–46]. Non-transformed seedlings were used as controls. After 3 days of inoculation, the disease symptoms were observed and photographed using a Nikon D700 camera. For pathogen level analysis, the relative accumulation of *P.sojae* in infected cotyledons and roots were also analyzed by qPCR using One Step RT-PCR Kit (Code No.:PCR-311, TOYOBO, Japan) on a QuantStudio 3 instrument (Thermo, United States). The expression of the *P. sojae* housekeeping gene *PsACT* (GenBank accession no. XM_009530461.1) relative to the soybean housekeeping gene *GmCYP2* (Glyma.12G024700.1) ($\Delta Ct = Ct_{HK \text{ of } P. sojae} - Ct_{HK \text{ of soybean}}$) was calculated. The housekeeping genes of *P. sojae* and soybean were chosen as described previously by Wang *et al.* (2011) [47], and the primers were listed in S1 Table.

## Detection of enzyme activities

To test whether *GmTCP19L* could affect superoxide dismutase (SOD) activity and peroxidase (POD) activity, the activities of SOD and POD were measured in *tcp19l* mutants one gram of

fresh roots were harvested at 3 days after inoculation with zoospores of *P. soja*e. The SOD and POD activities were measured as described previously by Li *et al.* (2015) [48]. Non-transformed seedlings were used as controls.

### Primer sequences used in the present study

The specific primers used for amplifying the regions which span the target sites, potential off-target sites, identifying *tcp19l* mutants without the BAR and Cas9 sequences of the CRISPR/Cas9 vector are listed in S1 Table.

### Statistics and reproducibility

All statistical methods are annotated in the figure captions. The numbers of biological replicates in each assays are also indicated in the figure captions. Three independent biological replicates were used for each sample and the student's *t*-test ($^*P<0.05$, $^{**}P<0.01$) was used to analyze statistical significance. Error bars represent ±SD.

## Results

### Isolation and phylogenic analysis of GmTCP19L

The full-length sequence of this *TCP* gene was isolated from "Williams 82" using RT-PCR technique. Alignment and phylogenetic tree analysis of the full-length amino acids sequence with the 24 TCP transcription factor members of Arabidopsis were performed. It showed the highest homolgy with AtTCP19 protein, and then this *TCP* gene was named as *GmTCP19-Like* (*GmTCP19L*) (S3 Fig).

### Targeted mutagenesis of *GmTCP19L* induced by CRISPR/Cas9

We designed sgRNAs within *GmTCP19L* gene using CRISPR-P, which displayed all optional sgRNA sequences (20 bp) immediately followed by 5'-NGG (PAM, protospacer adjacent motif) in the forward or reverse strand. Two target sites in the exon of *GmTCP19L* (named *GmTCP19L*-SP1 and *GmTCP19L*-SP2) were chosen (**Fig 1**), and the corresponding sgRNA/Cas9 vectors were transferred into the soybean cultivar "Williams 82" by *Agrobacterium*-mediated genetic transformation. The whole genome DNA was used to examine CRISPR/Cas9-induced mutations at the target sites using PCR and DNA sequencing analysis. In T0 transgenic lines, we determined that 15.7% (8 of 51) and 11.8% (6 of 51) DNA-positive plants had heterozygous mutations at the two target sites of *GmTCP19L*, respectively (**S4 Fig**). Then, all seeds

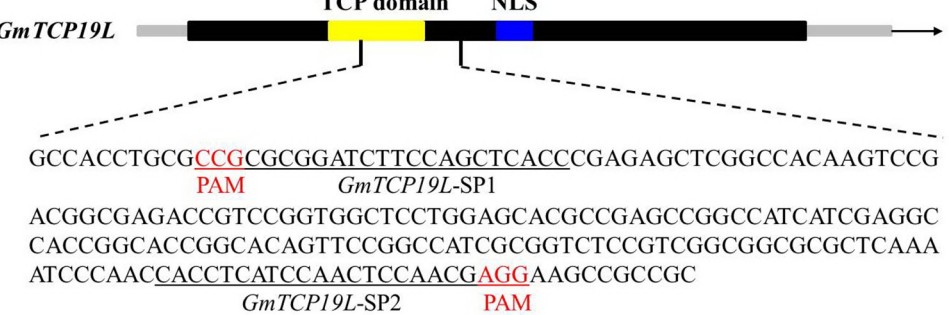

**Fig 1. Gene structures of *GmTCP19L* with target sites.** The underlined nucleotides indicate the target sequences, and the red nucleotides indicate PAM sequences.

**Table 1. CRISPR/Cas9-mediated targeted mutagenesis of *GmTCP19L* in the T1 generation.**

| T1 lines with *tcp19l* mutations | No. of plants sequenced | No. of homozygous *tcp19l* mutants | No. of heterozygous *tcp19l* mutants | No. of plants with no mutation |
|---|---|---|---|---|
| *tcp19l*-SP1-T1#01 | 26 | 0 | 18 | 8 |
| *tcp19l*-SP1-T1#02 | 18 | 9 | 7 | 2 |
| *tcp19l*-SP1-T1#07 | 26 | 7 | 16 | 3 |
| *tcp19l*-SP1-T1#08 | 12 | 0 | 7 | 5 |
| *tcp19l*-SP1-T1#10 | 12 | 0 | 12 | 0 |
| *tcp19l*-SP1-T1#20 | 44 | 0 | 32 | 12 |
| *tcp19l*-SP1-T1#21 | 77 | 13 | 43 | 21 |
| *tcp19l*-SP1-T1#51 | 11 | 0 | 8 | 3 |
| *tcp19l*-SP2-T1#02 | 18 | 5 | 7 | 6 |
| *tcp19l*-SP2-T1#07 | 26 | 8 | 17 | 1 |
| *tcp19l*-SP2-T1#10 | 12 | 0 | 6 | 6 |
| *tcp19l*-SP2-T1#20 | 44 | 0 | 26 | 18 |
| *tcp19l*-SP2-T1#21 | 77 | 0 | 28 | 49 |
| *tcp19l*-SP2-T1#51 | 11 | 2 | 9 | 0 |

derived from each T0 lines were planted under natural conditions. The types of mutation at target sites of *GmTCP19L* were observed in the T1 generation (**Table 1**). Sequencing analysis showed that a total of 44 T1 plants (29 *tcp19l*-SP1 and 15 *tcp19l*-SP2) were homozygous for null alleles of *GmTCP19L* and two types of mutations at target site *GmTCP19L*-SP1 (2-bp deletion and 1-bp deletion) were detected (**Fig 2A and 2B**). Meanwhile, the 14-bp deletion type of mutations was found at target site *GmTCP19L*-SP2 (**Fig 2A and 2C**). The frameshift mutations of three types at two target sites of *GmTCP19L* resulted in premature translation termination codons (**S5 Fig**).

## Potential off-target analysis

Then, we sought to determine whether introduction of CRISPR/Cas9 machinery led to off-target mutagenesis of soybean DNA, we identified the off-target sites in the "Williams 82" genome predicted to be most likely off-targets of our sgRNAs using the online tool CRISPR-P. Two most likely off-target sites with highest scores at the two target sites (*GmTCP19L*-SP1 and *GmTCP19L*-SP2) were examined by specific genomic PCR and DNA sequencing analysis in the 44 T1 plants identified as homozygous *tcp19l* mutants. Every potential off-target site only possessed mismatches of 2–4 bases compared with the *GmTCP19L* target sequences (**S2 Table**). In this study, no mutations were observed in the examined potential off-target sites (**Fig 3**, **S2 Table**).

## Phenotypes of T1 mutants in response to *P. sojae* infection

In the T1 generation, cotyledons of three types of homozygous *tcp19l* mutants (1-bp deletion and 2-bp deletion at target site *GmTCP19L*-SP1, 14-bp deletion at target site *GmTCP19L*-SP2) were selected to investigate resistance to *P. sojae*. We found that the cotyledons of *tcp19l* mutants became soft and exhibited clear water-soaked lesions compared with those of the WT after 3 days of incubation with zoospores of *P. sojae* (**Fig 4A**), and the accumulation of *P. sojae* in infected cotyledons was significantly (P<0.05) higher in *tcp19l* mutants (1-bp deletion and 2-bp deletion) than that in WT (**Fig 4B**). These results indicate that the cotyledons of T1 homozygous *tcp19l* mutants showed a reduced resistance phenotype after *P. sojae* infection.

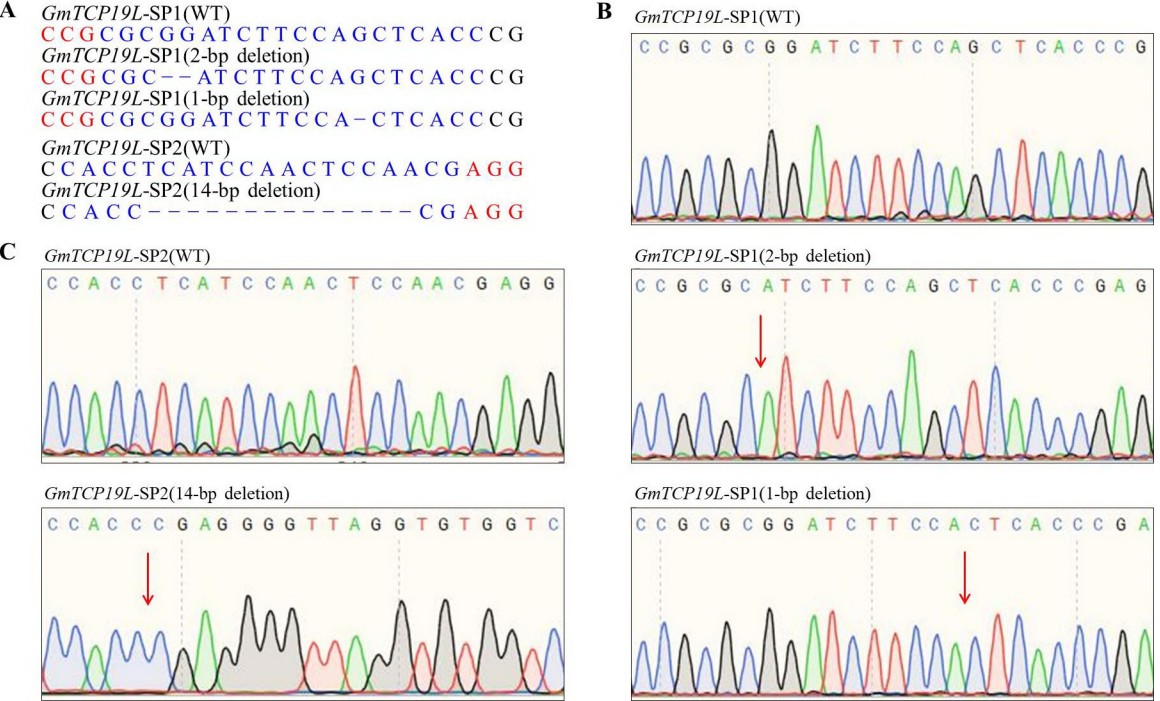

**Fig 2. Homozygous targeted mutagenesis of *GmTCP19L* induced by CRISPR/Cas9 in the T1 generation.** (A) Sequences of WT and representative mutation types induced at target sites *GmTCP19L*-SP1 and *GmTCP19L*-SP2. Dashes, deletions. (B) and (C) Sequence peaks of WT and representative mutation types at target sites *GmTCP19L*-SP1 and *GmTCP19L*-SP2. The red arrowheads indicate the location of mutations. WT, wild-type soybean plant.

## Generation of *tcp19l* mutants without the BAR and Cas9 sequences

To obtain soybean germplasm with homozygous *tcp19l* mutations but without the BAR and Cas9 sequences of the CRISPR/Cas9 vector, the selectable marker BAR was examined by test

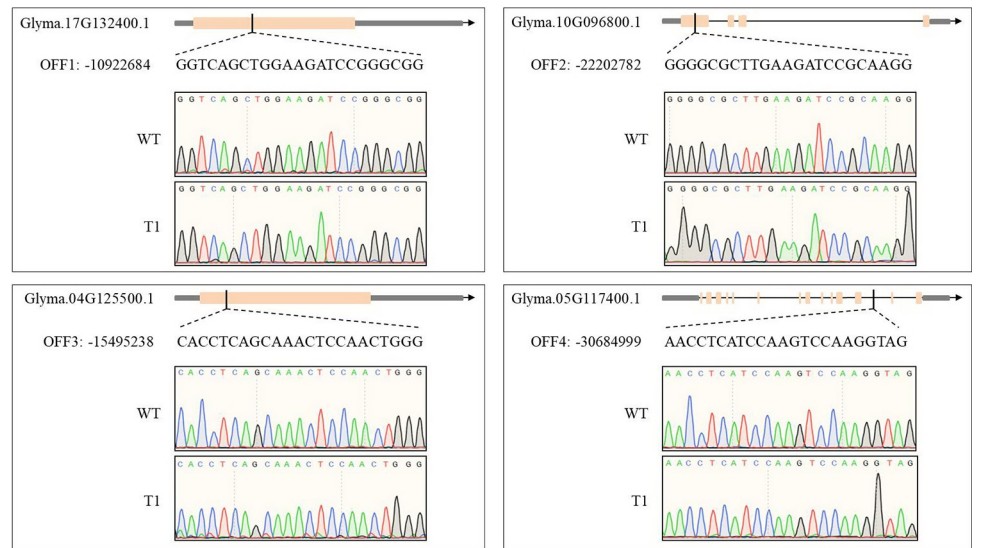

**Fig 3. Potential off-target analysis at the two target sites of *GmTCP19L* in the T1 generation.** WT, wild-type soybean plant. T1, the homozygous of *tcp19l* mutants in the T1 generation.

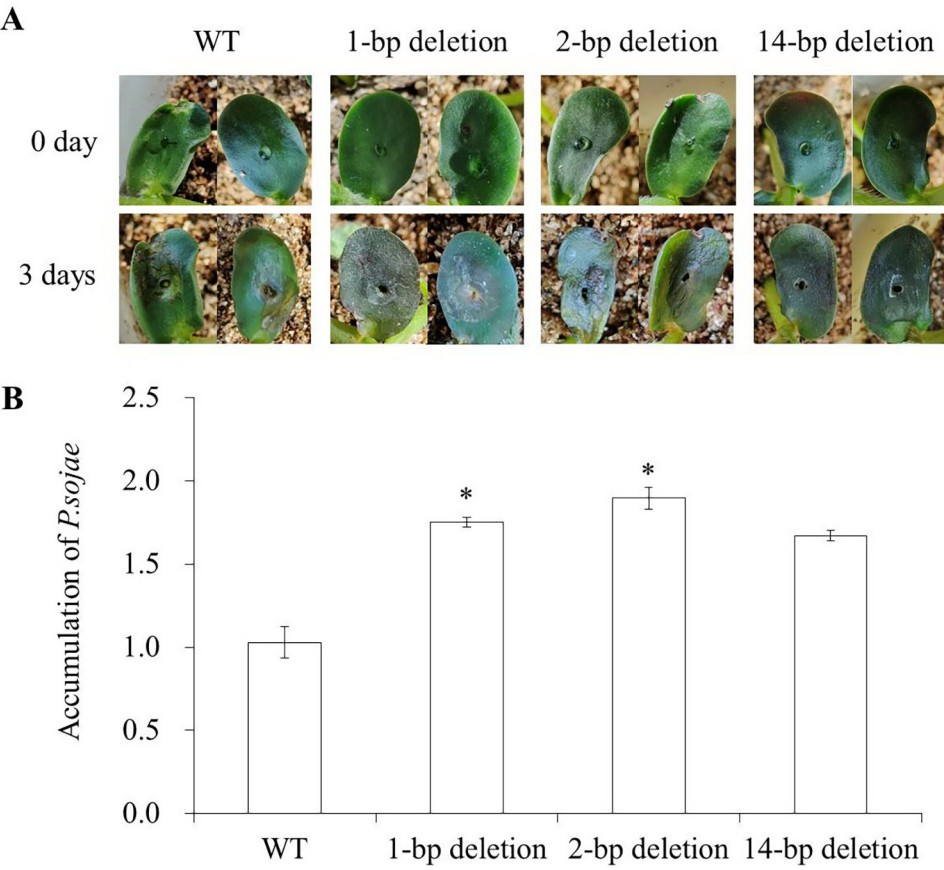

**Fig 4. Phenotypes of T1 mutants in response to *P. sojae* infection.** (**A**) Disease symptoms on the cotyledons of T1 *tcp19l* mutants and WT treated with *P. sojae* after 3 days. (**B**) Accumulation of *P. sojae* in *tcp19l* mutants and WT. Three independent biological replicates were used for each sample and the student's *t*-test (*$P<0.05$) was used to analyze statistical significance. Error bars represent ±SD.

PAT/Bar strip, and the Cas9 coding sequence were amplified by PCR technique. In T1 generation, we found that two of 44 homozygous *tcp19l* mutants were both BAR-free and *Cas9*-free (**Fig 5A and 5B**). Then, these two *tcp19l* mutants were named as *tcp19l*-SP1-T1#02.03 and *tcp19l*-SP1-T1#02.08. Simultaneously, the 2-bp deletion type of mutations at target site *GmTCP19L*-SP1 was found in both *tcp19l*-SP1-T1#02.03 and *tcp19l*-SP1-T1#02.08, and the sequence of them were consistent (**S6 Fig**).

## Phenotypes of the progeny of *tcp19l* mutants in response to *P. sojae* infection

In the T2 generation, the progeny of T1 homozygous *tcp19l* mutants *tcp19l*-SP1-T1#02.03 and *tcp19l*-SP1-T1#02.08 were grown under natural conditions. Because of the mutant of them were consistent, we number them uniformly as *tcp19l*-SP1-T2, and all of them were homozygous *tcp19l* mutants without the BAR and Cas9 sequences (**Table 2**). To examine the phenotype of *tcp19l*-SP1-T2 in response to *P. sojae* infection, eight plants were selected to investigate resistance to *P. sojae* by the wounded-hypocotyl inoculation technique. After 3 days of incubation, a remarkable difference in the development of disease symptoms was observed. The hypocotyls of *tcp19l*-SP1-T2 became soft, exhibited clear water-soaked lesions and emitted a foul odor compared with those of the WT, and there were nearly no visible lesions in WT and

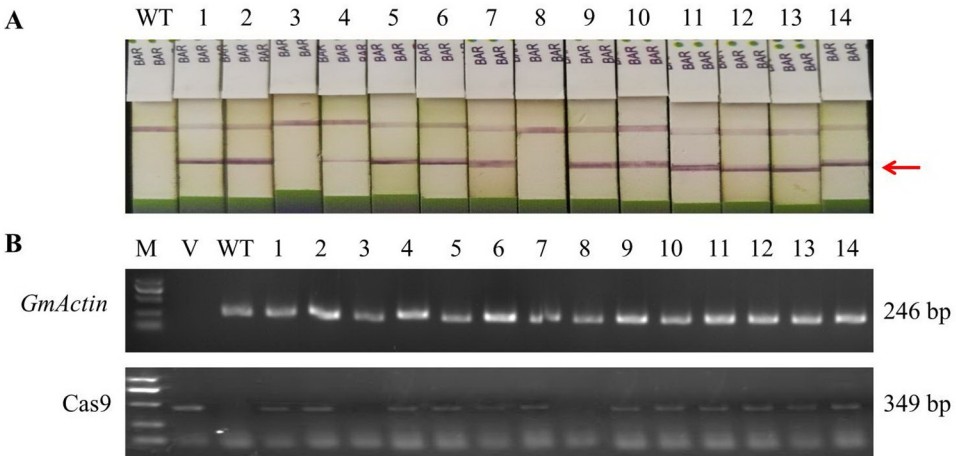

**Fig 5. The partial result of identifying *tcp19l* mutants without the BAR and Cas9 sequences of the CRISPR/Cas9 vector. (A)** Detection of the selectable marker BAR by test strip. WT, wild type soybean plant. Labels 1–14, individual mutant plants. The bands at red arrowhead indicate that BAR is positive. **(B)** Gel images of PCR products obtained with a set of primer pairs for the Cas9 of sgRNA/Cas9 vectors. Cas9 (349 bp), part of the Cas9 coding sequence. *GmActin* (Glyma.18G290800.1) was used as a normalization control. M, DL2000 DNA marker. V, plasmid of CRISPR/Cas9 vector used in transformation. WT, DNA of wild type soybean plant. Labels 1–14, individual mutant plants.

the hypocotyls keep hard (**Fig 6**). In the T3 generation, similar results were obtained at 3 days after inoculation with zoospores of *P. sojae* in the soybean root hydroponic assay, the disease symptom of root browning and stem stunting appeared in *tcp19l* mutants. In contrast, the WT seedlings were clearly healthier than *tcp19l* mutants seedlings (**Fig 7A**). We also analysed the accumulation of *P. sojae* in infected living roots after 3 days of incubation with *P. sojae* zoospores. The accumulation of *P. sojae* was significantly (P<0.01) higher in *tcp19l* mutants than that in WT (**Fig 7B**). Meanwhile, the activities of SOD and POD were significantly (P<0.01) decreased in *tcp19l* mutants compared with WT seedlings (**Fig 7C and 7D**). These results indicate that the susceptibility to *P. sojae* was enhanced in *tcp19l* mutants.

## Discussion

Soybean [*Glycine max* (L.) Merr.] is an important food crop with great economic value that abundant protein and oil. Phytophthora root and stem rot, which is caused by the oomycete *P. sojae* has been observed in all major soybean-growing regions all around the world [49–51]. Because Phytophthora root and stem rot is a persistent problem for soybean, improvement of disease resistance through breeding, biocontrol and biotechnology approaches are all active areas of research. Therefore, exploring the molecular mechanisms involved in responses to *P. sojae* infection can provide useful information to generate resistant cultivars through molecular breeding.

With the development of the CRISPR/Cas9 system as a tool for targeted genome editing [52], it quickly became an effective method for crop improvement [53–55]. The system has

**Table 2. Identifying *tcp19l* mutants without the BAR and Cas9 sequences from T1 and T2 generations.**

| T1 homozygous *tcp19l* mutants | T-DNA in the T1 homozygous *tcp19l* mutants | No. of the progeny plants identified | No. of the T2 homozygous *tcp19l* mutants without the BAR and Cas9 sequences |
|---|---|---|---|
| *tcp19l*-SP1-T1#02.03 | T-DNA-free | 16 | 16 |
| *tcp19l*-SP1-T1#02.08 | T-DNA-free | 15 | 15 |

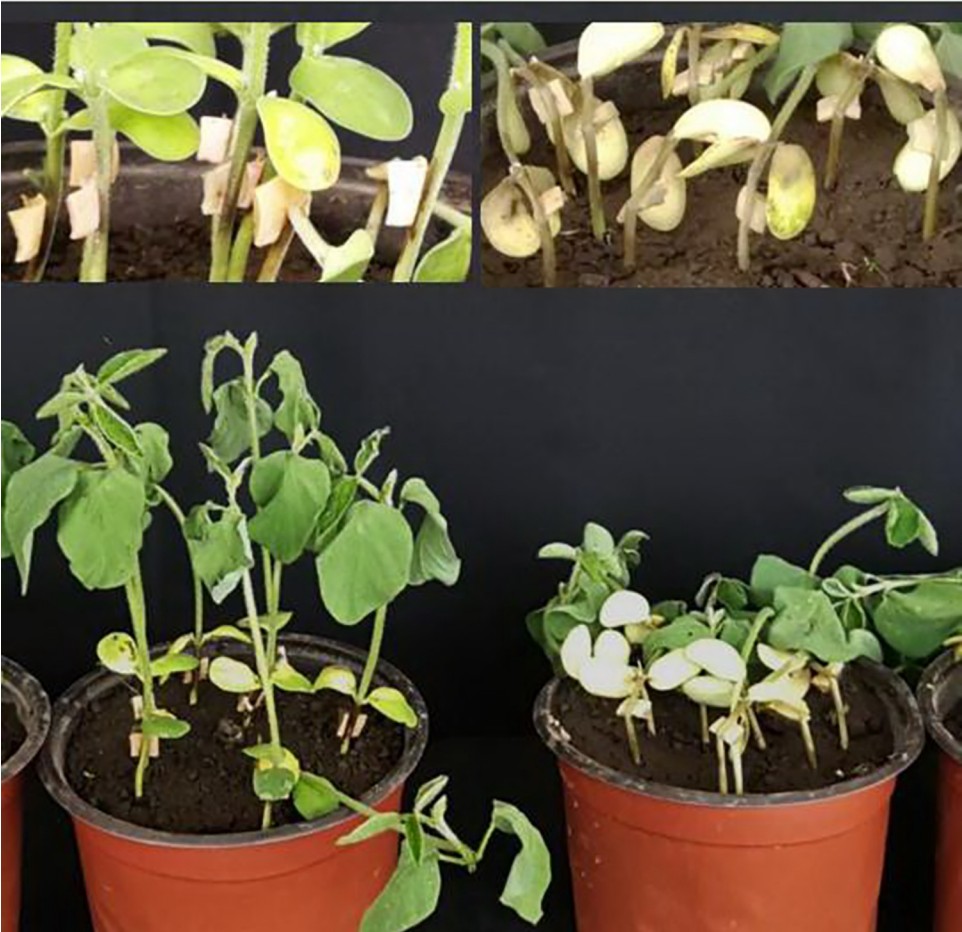

**Fig 6. Disease symptoms on the hypocotyls of T2 *tcp19l* mutants and WT treated with *P. sojae* after 3 days.** WT, wild-type soybean plant. *tcp19l*-SP1-T2, 2-bp deletion mutants, the progeny of T1 *tcp19l* mutants *tcp19l*-SP1-T1#02.03 and *tcp19l*-SP1-T1#02.08.

already been widely used in many crops and there are also many applications in soybean. Cai et al. (2018) employed the CRISPR/Cas9 system to specifically induce targeted mutagenesis of the *GmFT2a* gene in soybean, and the homozygous GmFT2a mutants exhibited late flowering under both long-day and short-day conditions [56]. Subsequently, the CRISPR/Cas9 system was also used to target four *GmLHY* genes in soybean, and the height and internodes of the *GmLHY* mutants were significantly shorter than that of the WT [57]. Han et al. (2019) obtained *E1* gene mutants using the CRISPR/Cas9 system and *Agrobacterium*-mediated transformation technique, and the mutants exhibited obvious early flowering under long day condition [58]. In this study, we screened and obtained homozygous mutants without the BAR and Cas9 sequences targeting soybean endogenous gene *GmTCP19L* using the CRISPR/Cas9 system, which will significantly increase breeding efficiency and speed up breeding process. GmTCP19L is the first *TCP* gene was identified in soybean response to *P. sojae* infection. In previous studies, TCP transcription factors have been shown to be a versatile regulatory role at

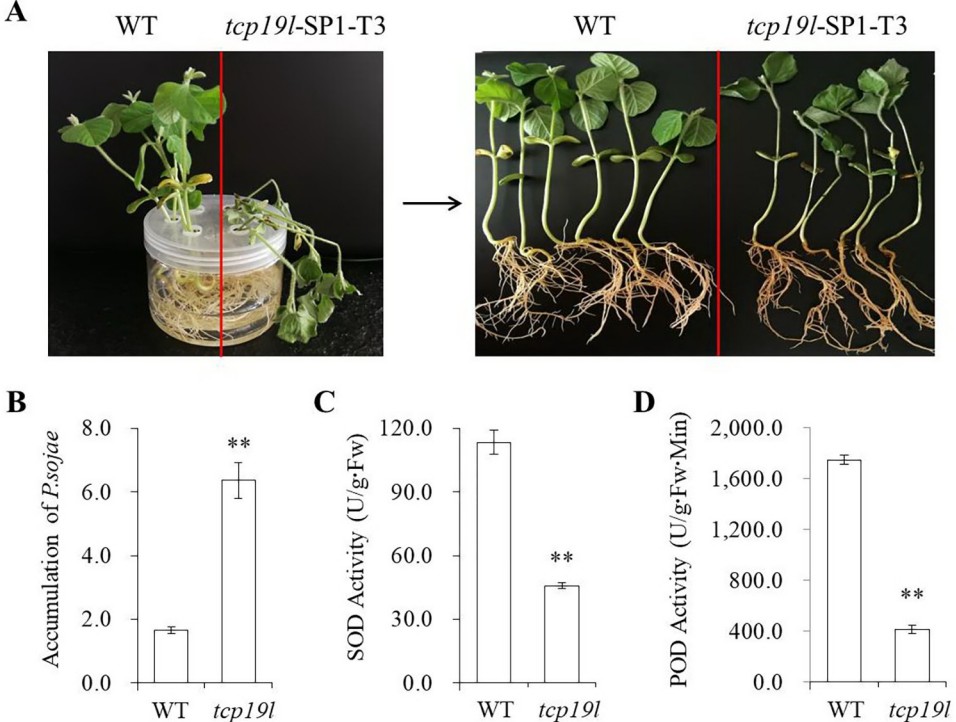

**Fig 7. Analysis of the *tcp19l*-SP1-T3 mutants and WT treated with zoospores of *P. sojae* in the soybean root hydroponic assay. (A)** Disease symptoms on the roots of *tcp19l* mutants and WT treated with zoospores of *P. sojae* at 3 days after inoculation. **(B)** Accumulation of *P. sojae* in *tcp19l* mutants and WT. **(C, D)** The activities of SOD and POD in *tcp19l* mutants and WT at 3 days after *P. sojae* inoculation. Three independent biological replicates were used for each sample and the student's *t*-test (*$P<0.05$, **$P<0.01$) was used to analyze statistical significance. Error bars represent ±SD.

the different stage of plant growth and development, such as leaf development [37], leaf morphogenesis [8], leaf senescence [20], trichome development [30, 59], flower development [19, 60, 61], circadian clock [62], hormone signaling [28, 63, 64], and in the response to varies stresses [35, 64–67]. Recently, increasing experimental evidence has showed that TCP transcription factors played a pivotal role in plant defense [22, 33]. In Arabidopsis, the *tcp13*, *tcp14* and *tcp19* single mutants display enhanced disease susceptibility to 2 different avirulent *Hyaloperonospora arabidopsidis* (*Hpa*) isolates (Emwa1 and Emoy2), while the *tcp15* mutant exhibits improved disease resistance to the virulent *Hpa* isolate Noco2 [35]. AtTCP8, AtTCP14, and AtTCP15 physically interact with NPR1 and function redundantly to establish systemic acquired resistance (SAR), and AtTCP15 promotes the expression of *PR5* which belongs to SAR marker genes [68]. Silencing of *StTCP23* in potato causes stunting and a branched phenotype as well as increasing susceptibility to common scab disease caused by the bacterial pathogen *Streptomyces turgidiscabies* [69]. Previous studies found that the *GmTCP19L* was upregulated in several near isogenic lines (NILs) under *P. sojae* treatment as assessed by RNA-Seq, but little is known about the functional roles in soybean [38]. Consistent with this, in this study, we further demonstrated that the accumulation of *P. sojae* was significantly higher in *tcp19l* mutants than that in WT. These results indicate that *GmTCP19L* is required for soybean defense responses to *P. sojae*.

Additionally, to survive the damaging effect of stresses-induced ROS accumulation, plants have evolved multifaceted strategies to minimize these adverse effects. Protective enzymes in

plant organisms, such as POD and SOD, can reduce ROS levels by scavenging free radicals hydrogen peroxide ($H_2O_2$) or superoxide ($O^{2-}$), leading to improve the plant resistance against pathogens [70, 71]. GmPIB1 (a bHLH-type transcription factor) has been shown to facilitate resistance to *P. sojae* in *Glycine max* by affecting ROS levels [72]. In this study, the activities of SOD and POD were significantly lower in *tcp19l* mutants after *P. sojae* infection than those in the WT plants, suggesting that *GmTCP19L* may improve the resistance by regulating the anti-oxidant defense system.

Plants must fine-tune defense responses to avoid deleterious effects on growth and development. In previous studies, there were reports that a few of TCP genes also play a wide range of roles in growth and development [8, 19, 20, 28, 30, 35, 37, 59–67]. But, we did not find any other phenotypic changes from T0, T1, T2, or T3 *tcp19l* mutants. So, in the follow-up work, we will continue to multiply the number of seeds, and it is valuable that exploring enough phenotypes of the *tcp19l* mutants and network involved in regulation of *GmTCP19L*, it will improve the interpretation of the findings that the function of the *GmTCP19L* response to varies stresses. Our work provides a new soybean germplasm with homozygous *tcp19l* mutations but the BAR and Cas9 sequences were undetectable using strip and PCR methods, respectively, suggesting directions for the breeding or genetic engineering of disease-resistant soybean plants.

## Supporting information

**S1 Fig. The sequences of the Cas9 and *Glycine max U6* promoter used in the present study.** (PDF)

**S2 Fig. The basic architecture of the constructs used for *GmTCP19L*-CRISPR/Cas9-mediated genome editing.** GmU6, *Glycine max U6* promoter. sgRNA, small guide-RNA. *GmTCP19L*-SP1/SP2, two target sites in the exon of *GmTCP19L*. Cas9-F/R, the primers of the detection region for Cas9. NLS, nuclear localization sequence. The *bar* gene driven by a CaMV 35S promoter is used as a screening marker. Kan, kanamycin resistance gene. pVS1, pVS1 replication origin. STA, pVS1 stability function. (PDF)

**S3 Fig. Phylogenetic tree analysis of GmTCP19L with the 24 TCP transcription factor members of Arabidopsis. (A)** The PCR amplified products of *GmTCP19L*. M, DL2000 DNA Marker. **(B)** Phylogenetic tree analysis of GmTCP19L with the 24 TCP transcription factor members of Arabidopsis. (PDF)

**S4 Fig. Heterozygous targeted mutagenesis of *GmTCP19L* induced by CRISPR/Cas9 in the T0 generation.** WT, wild-type soybean plant. (PDF)

**S5 Fig. Amino acid sequence alignment of *GmTCP19L* mutations with WT.** The TCP-like domain was marked with a black box. Nuclear location signal was marked with a dashed black box. (PDF)

**S6 Fig. Frameshift mutations at two target sites of *GmTCP19L* generated premature translation termination codons.** CDS, coding sequence. Blue capital letter, target sequence. Red capital letter, protospacer adjacent motif. Dashes, deletions. Yellow rectangle, termination codon. (PDF)

**S1 Table. Primer sequences used in the present study.**
(PDF)

**S2 Table. Potential off-target analysis at the two target sites of *GmTCP19L* in the T1 generation.**
(PDF)

**S1 Raw images.**
(PDF)

**S1 Data.**
(ZIP)

**S1 File.**
(DOCX)

## Acknowledgments

We would like to thank Shuzhen Zhang and her team (Soybean Research Institute, Key Laboratory of Soybean Biology of Chinese Education Ministry, Northeast Agricultural University, Harbin, China) for providing the *Phytophthora sojae* race 1 strain used in this study.

## Author Contributions

**Conceptualization:** Sujie Fan, Jun Zhang, Piwu Wang.

**Data curation:** Sujie Fan, Yang Song, Piwu Wang.

**Formal analysis:** Sujie Fan, Jun Zhang, Piwu Wang.

**Funding acquisition:** Sujie Fan, Zhuo Zhang, Yang Song.

**Investigation:** Sujie Fan, Zhuo Zhang, Yang Song.

**Methodology:** Sujie Fan, Zhuo Zhang, Yang Song.

**Project administration:** Sujie Fan, Jun Zhang, Piwu Wang.

**Resources:** Sujie Fan, Zhuo Zhang, Jun Zhang, Piwu Wang.

**Software:** Jun Zhang.

**Supervision:** Zhuo Zhang, Yang Song, Jun Zhang, Piwu Wang.

**Validation:** Sujie Fan, Zhuo Zhang, Yang Song.

**Visualization:** Jun Zhang.

**Writing – original draft:** Sujie Fan.

**Writing – review & editing:** Zhuo Zhang, Yang Song, Jun Zhang, Piwu Wang.

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
