## [Decision Letter · Decision Letter 0]

8 Nov 2021

PONE-D-21-31763CRISPR/Cas9-mediated targeted mutagenesis of GmTRP19 increasing susceptibility to Phytophthora sojae in soybeanPLOS ONE

Dear Dr. Fan,

Thank you for submitting your manuscript to PLOS ONE. After careful consideration, we feel that this study has scientific merit but the current manuscript does not fully meet PLOS ONE’s publication criteria. The reviewers pointed out several aspects such as the writing, statistics, and concerns about CRISPR specificity and additional phenotypes that need to be further clarified. Therefore, we invite you to submit a revised version of the manuscript that addresses the points raised during the review process.

We look forward to receiving your revised manuscript.

Kind regards,

Hao-Xun Chang, Ph.D.

Academic Editor

PLOS ONE

Journal Requirements:

Reviewers' comments:

Reviewer's Responses to Questions

**Comments to the Author**

1. Is the manuscript technically sound, and do the data support the conclusions?

Reviewer #1: Partly

Reviewer #2: Yes

Reviewer #3: Yes

2. Has the statistical analysis been performed appropriately and rigorously? 

Reviewer #1: N/A

Reviewer #2: N/A

Reviewer #3: No

5. Review Comments to the Author

**Reviewer #1:** 1. The authors used CRISPR/Cas9 based genome editing to knock out a soybean TCP domain-containing gene Glyma05g03610 to study its function. They designed a genome editing DNA construct targeting two sites around the critical TCP domain of the gene, transformed Williams 82 and obtained up to 14 T1 lines with either of the two sites edited (according to Table 1). But they either failed to characterize all the edited lines in detail or did not report the results. For example, the chromatographs in S3 Fig can be easily replaced with conclusive sequences of the edited events by sequencing a few clones of the PCR fragments amplified from the target region. They can also carry the heterozygous events to T2 generation to obtain transgene-free homozygous mutants to study their phenotypical effects. In other words, the authors achieved the most important part of the study by obtaining multiple edited events but failed to do follow-up study on them to get more comprehensive results.

2. Although the authors summarized in the Introduction and Discussion that TCP transcription factors can function in various aspects such as leaf development [37], leaf morphogenesis [2], leaf senescence [20], trichome development [30, 55], flower development [13, 56, 57], circadian clock [58], hormone signaling [28, 59, 60], response to varies stresses [35, 60–63], and plant defense [22, 33], they assumed the Glyma05g03610 gene in this played a role in plant defense and only did a simple infection experiment using only one mutant line. With as many as 14 edited lines in hands, the authors should evaluate all aspects of the edited plants and compare multiple lines at multiple stages to wild type controls to get a more convincing conclusion about the function of the gene. If possible, they should also get molecular and/or biochemical evidence to support their conclusion.

3. What is the “R” in GmTRP19 standing for since all the other related genes are named as TCP. It was first described on page 3 lines 8-11 “In a previous study, GmTRP19 (Glyma05g03610, a TCP transcription factor related to P. sojae) was significantly induced after infection with P. sojae among several soybean near isogenic lines revealed by comparative transcriptomics [38].” But it seems that neither the term of GmTRP19 or Glyma05g03610 is listed in the reference.

4. The manuscript should be carefully checked and edited before submitting. There are numerous errors or improper use of English such as those listed below.

Page 1 line 14: transformation were used

Page 2 line 4: causing obvious increasing

Page 2 line 4: transgenic element of the CRISPR/Cas9 vector, which suggesting directions for

Page 2 lines 12-18 should be placed after the general introduction of TCP on page 3. The entire Introduction should be better organized.

Page 4 line 9: auxin-induced related genes

Page 5 line 6: expression of LOX2 gene and reduced the production of JA,

Page 5 line 13: function of this gene resistance to P. sojae.

Page 5 line 15: and maintained consistent mutation types

Page 5 line 16: A certain amount of 2-bp deletion trp19 mutants

Page 6 line 11: P. sojae race 1 was kindly provided by professor zhang?

Page 7 line 1: Having demonstrated that the

Page 7 line 6: the Glycine max U6 promoter (GmU6) within its T-DNA region: The Cas9 is not expressed by the U6 promoter.

Page 7 line 7: The bar/basta gene: bar is the Phosphinothricin N-acetyltransferase gene of Streptomyces hygroscopicus. Phosphinothricin is the active component of commercial herbicide Basta.

Page 7 line 10: The GmTRP19-CRISPR/Cas9 positive plasmids

Page 8 line 5: Simultaneously, we sought to: Transgene-free mutants can only be produced in T1 or T2 progenies through genetic segregation.

Page 8 line 10: potential off-target sites effecting on phenotype,

Page 8 line 12: off-target sites for each guide would be selected,

Page 8 lines 15-16: to determine whether there were presence of the off-target sites in the amplicon by sequenced.

Page 8 line 18: The “transgene-clean” trp19 mutant lines: Transgene-free

Page 9 line 6: the significance differences

Page 10 line 2: targeted PAM in the exon of GmTRP19 (Fig 1). The CRISPR/Cas9 construct were

Page 10 lines 12-13: homozygous for null alleles of GmTRP19 and two types of mutations at target site GmTRP19-SP1 (2-bp deletion and 1-bp deletion) were detected (Fig 2A, B).: The description is incomplete and not clear. There were homozygous mutant plants detected in three trp19-SP1 T1 lines and also in three trp19-SP2 T1 lines according to Table 1. A total of 14 T1 lines had heterozygous mutant plants. What happened in them? Indels in the target sequences of these homozygous mutant plants should be listed. Indels in the target sequences of the heterozygous mutant plants should also be experimentally determined and then listed.

Page 12 lines 18-19: and the part of Cas9 coding sequence in the T-DNA of CRISPR/Cas9 vector was also spanned.

Page 13 line 8: type soybean plant. Labels 1–14, individual mutant lines. Should “lines” be plants?

Page 13 line 10: T-DNA regions of CRISPR/Cas9 vector.

Page 13 line 17: and all of their

Page 15 line 2: is an important food and economic crops

Page 15 line 17: shortened internodes use of the CRISPR/Cas9 system in soybean

Page 15 line 18: obtained E1 gene mutants using CRISPR/Cas9 and soybean transformation,

Page 16 lines 1-2: GmTRP19 by CRISPR/Cas9 ge 1 nome editing approach without any transgenic element of the CRISPR/Cas9 vector, which quickly became a powerful resource for soybean

Page 16 line 5: Having demonstrated that

Page 16 line 8: which suggesting directions for the breeding or genetic engineering of

Page 30 line 4: S2 Fig. The basic architecture of the constructs

Page 30 line 6: promoter. sgRNA, small guide-RNA.

S5 Fig. The trp19-SP1-T1#02.03 (2-bp deletion) and trp19-SP1-T1#02.08 (2-bp deletion) are the same sequence since the two plants are derived from the same T1 line. No need to list the same sequence twice. Where is the (1-bp deletion) line or the 14-bp deletion line?

S1 Table. GmActin primers?

**Reviewer #2:** Manuscript Number: PONE-D-21-31763

Full Title: CRISPR/Cas9-mediated targeted mutagenesis of GmTRP19 increasing susceptibility to Phytophthora sojae in soybean.

Opinion:

The article proposed to study the function of the GmTRP19 gene in soybean plants, which is expected to be involved in defense against Phytophthora sojae.

For this, they used the CRISPR/Cas9 gene editing system to cause an interruption in the expression of GmTRP19 gene.

Plants with the edited gene showed an increase in susceptibility to the phytopathogen. So they concluded that directly or indirectly this gene regulates the resistance to P. sojae in soybean.

The article also shows the usefulness of the CRISPR system to generate edited plants, nonetheless non-GMO, which avoids the long and costly process of deregulation.

It would be interesting to explore the possible positive effect of the expression of that gene in soybean plants, and instead of interrupting its expression, as done here, increase its expression, and thus obtain a product with an important agronomic characteristic, and at the same time corroborate with the answer obtained at this stage of the work.

The article must be published as is.

**Reviewer #3: **The research article titled “CRISPR/Cas9-mediated targeted mutagenesis of GmTRP19 increasing susceptibility to Phytophthora sojae in soybean” explored the role of a plant-specific transcription factor in soybean disease resistance to an important crop pathogen, Phytophthora sojae. The manuscript contains a number of grammatical errors, though the content and concepts presented by the authors are logical and easy to follow. In addition, the molecular work is convincing that the CRISPR/Cas9 constructs successfully created mutations in their target gene, that some lines were homozygous for the mutations, and that some lines also lacked the genetic markers and can accurately be described as “transgene-free” mutants. Unfortunately, I have multiple concerns that prevent the manuscript from being acceptable for publication at this time.

First, there is no data presented to support the claim that there were no off-target effects from their CRISPR/Cas9 activity. The authors claim there are no off-target effects, yet only provide a supplementary table listing “number of plants with mutations” in off target sites, filled with zeros. The authors describe using the CRISPR-P analysis program, which provides scores based on strength of sequence alignment with a CRISPR construct. The authors ought to describe what threshold they used to identify putative “off-target” candidates, and they also ought to provide some chromatograms to show that these putative off-target sites were not mutated by the CRISPR constructs.

Second, I don’t think the authors explore enough phenotypes of their mutants. The authors acknowledge at many points in the manuscript that this class of transcription factor play a wide range of roles in growth, development, flowering, circadian rhythm, etc. But, the authors only present one phenotype – susceptibility to 1 isolate of P. sojae. Therefore, the increased susceptibility to P. sojae could be one of many phenotypes of these mutant lines, which could compromise the conclusion that GmTRP19 has a direct role in resistance to P. sojae. Additionally, the phenotype that the authors do test are only observational, with no numeric data or statistics to suppor the conclusion of increased susceptibility in the mutants.

If the authors address these concerns, it could be re-considered for publication in PLOS One.

Specific Comments:

Page 2, Line 15: Remove “the” before “conserved domains”

Page 4, Line 3: What is the FT-FD complex. Please list in full before using an abbreviation.

Page 4, Line 4: What is the AP1 gene, and what is the relevance of AtTCP12 and AtTCP18 binding to it?

Page 4, Line 17: Hyaloperonospora arabidopsidis is not a gram-negative bacteria. It is an oomycete. Please correct.

Page 5, Line 9: Remove “related to P. sojae”. You describe the connection between the gene and P. sojae later in the sentence.

Page 5, Line 18: Remove “the” before “resistance”. There is no singular resistance to P. sojae – there is single, R-gene mediated resistance, and partial resistance (Dorrance, 2018 doi.org/10.1080/07060661.2018.1445127). The GmTPS19 gene is not an R-gene.

Page 6, line 6: Williams 82 has high partial resistance to P. sojae, but only 1 R-gene mediating robust resistance to specific pathotypes of P. sojae (Dorrance, 2018).

Page 6, line 11: Capitalize “Professor Zhang”.

Page 8, line 6: What is a PAT/Bar “test strip”? Did you spray these plants with gufosinate? Did you do PCR to detect the presence / absence of this gene? Did you get strips from a company? If so, where? More detail needed here for reproducibility.

Page 8, line 12: “Two off-target sites for each guide would be selected” does this mean there were 2 off target sites detected by CRISPR-P analysis? Or only 2 were selected for analysis of off-target effects? There’s a big difference, please clarify.

Page 10, lines 7-8: Please clarify the language here a bit. You list two numbers and use “respectively”, but don’t list both conditions. Please change to something like this “…had heterozygous mutations at the first (SP1) and second (SP2) target sites, respectively (S3 Fig in S1 File)”.

Page 10, line 16: Please don’t use “PTCs” as an abbreviation. This is unnecessary.

Page 12, lines 8-9: How “likely” were these two off-target sites predicted to be affected? How many other off-target sites were identified at a lower “likelihood”? Can you provide the results of the CRISPR-P analysis in the supplemental files, or list here in some form?

Page 12, lines 12-13: “no mutations were observed” yet the data to support this is just a table with 0’s listed. Please provide the chromatograms to show evidence that there were no off-target effects on DNA sequence.

Page 13, Line 17: “all of their…” all of their what? I think there is a word missing here. Or perhaps “their” is meant to be “them”?

Figure 4: These soybeans do not look like they are growing in vermiculite, as described throughout the manuscript. Explain.

Page 15, lines 15-19: This brings up a great point. Did any of your GmTRP19 mutants display any other phenotypes besides increased sensitivity to P. sojae? Were there any developmental phenotypes (delayed germinating, effects on plant height, flowering, etc.)?

Page 16, line 4: What makes GmTRP19 a “new” member of this family?

Page 16, lines 5-9: This entire sentence is an incomplete sentence. Authors could remove “Having demonstrated that”, to make it complete.

A few examples of improper grammar. Many others throughout. PLOS One does not copyedit accepted manuscripts.

• “We obtained a novel type of mutations”

• “which suggesting”

• “this gene resistance to P. sojae”

• “exhibited more sensitive to P. sojae”

• “All of three types”

• “an important food and economic crops”

---

## [Author Response · Author response to Decision Letter 0]

20 Jan 2022

Response to Reviewer#1:

Q1: The authors used CRISPR/Cas9 based genome editing to knock out a soybean TCP domain-containing gene Glyma05g03610 to study its function. They designed a genome editing DNA construct targeting two sites around the critical TCP domain of the gene, transformed Williams 82 and obtained up to 14 T1 lines with either of the two sites edited (according to Table 1). But they either failed to characterize all the edited lines in detail or did not report the results. For example, the chromatographs in S3 Fig can be easily replaced with conclusive sequences of the edited events by sequencing a few clones of the PCR fragments amplified from the target region. They can also carry the heterozygous events to T2 generation to obtain transgene-free homozygous mutants to study their phenotypical effects.In other words, the authors achieved the most important part of the study by obtaining multiple edited events but failed to do follow-up study on them to get more comprehensive results.

Thanks very much for your valuable comments. In low-generation, the number of homozygous plants obtained by screening is small, and one of the most important purposes of this research is to obtain new germplasm materials for breeding research, in this case, in order to prevent death of the mutant plants after P. sojae infection, the transgene-free mutants were examined the phenotype response to P. sojae infection. In addition, in the follow-up work, it is valuable that exploring enough phenotypes of the mutants and network involved in regulation of GmTCP19L, it will improve the interpretation of the findings that the molecular mechanism of the GmTCP19L response to varies stresses.. In the revised manuscript, we supplemented this relevant content in the discussion section.

Q2: Although the authors summarized in the Introduction and Discussion that TCP transcription factors can function in various aspects such as leaf development [37], leaf morphogenesis [2], leaf senescence [20], trichome development [30, 55], flower development [13, 56, 57], circadian clock [58], hormone signaling [28, 59, 60], response to varies stresses [35, 60–63], and plant defense [22, 33], they assumed the Glyma05g03610 gene in this played a role in plant defense and only did a simple infection experiment using only one mutant line. With as many as 14 edited lines in hands, the authors should evaluate all aspects of the edited plants and compare multiple lines at multiple stages to wild type controls to get a more convincing conclusion about the function of the gene. If possible, they should also get molecular and/or biochemical evidence to support their conclusion.

Thanks very much for your valuable comments. We added the molecular and biochemical evidence to support our conclusion in the revised manuscript: For disease resistance analysis, the soybean root hydroponic assay was performed in the T3 generation, the disease symptom of root browning and stem stunting appeared in tcp19l mutants at 3 days after inoculation with zoospores of P. sojae, and the WT seedlings were clearly healthier than tcp19l mutants seedlings (Fig 6A). We also analysed the accumulation of P. sojae in infected living roots after 3 days of incubation with P. sojae zoospores. The accumulation of P. sojae was significantly (P<0.01) higher in tcp19l mutants than that in WT (Fig 6B). Meanwhile, the activities of SOD and POD were significantly (P<0.01) decreased in tcp19l mutants compared with WT seedlings (Fig 5C, D). These results indicate that the susceptibility to P. sojae was enhanced in tcp19l mutants.

Q3: What is the “R” in GmTRP19 standing for since all the other related genes are named as TCP. It was first described on page 3 lines 8-11 “In a previous study, GmTRP19 (Glyma05g03610, a TCP transcription factor related to P. sojae) was significantly induced after infection with P. sojae among several soybean near isogenic lines revealed by comparative transcriptomics [38].” But it seems that neither the term of GmTRP19 or Glyma05g03610 is listed in the reference.

In the reference 38, a TCP gene (Glyma05g03610) was significantly induced after infection with P. sojae among several soybean near isogenic lines revealed by comparative transcriptomics. This TCP gene was listed in the transcriptome data of the additional files. In order to name this gene more accurately, alignment and phylogenetic tree analysis of the full-length amino acids sequence with the 24 TCP transcription factor members of Arabidopsis were performed. It showed the highest homolgy with AtTCP19 protein, and then this TCP gene was renamed as GmTCP19-Like (GmTCP19L) (S3 Fig in S1 File).

Q4: Specific comments:

1. Page 1 line 14: transformation were used

We have replaced “transformation were used” with “transformation was used”.

2. Page 2 line 4: causing obvious increasing

We have rephrased this sentences as “increasing susceptibility to P. sojae infection in the T2-generation”.

3. Page 2 line 4: transgenic element of the CRISPR/Cas9 vector, which suggesting directions for

We have removed the word “which”.

4. Page 2 lines 12-18 should be placed after the general introduction of TCP on page 3. The entire Introduction should be better organized.

We have adjusted it based on your suggestions.

5. Page 4 line 9: auxin-induced related genes

We have replaced “auxin-induced related genes” with “auxin-induced genes”.

6. Page 5 line 6: expression of LOX2 gene and reduced the production of JA,

We have rephrased this sentences as “inhibiting the expression of LOX2 gene and reducing the production of JA, and improving the reproduction ability of insects”.

7. Page 5 line 13: function of this gene resistance to P. sojae.

We have rephrased this sentences as “studied the function of this gene responses to P. sojae infection”.

8. Page 5 line 15: and maintained consistent mutation types

We have rephrased this sentences as “the targeted mutations were stably inherited from the T1 to T2 generation”. 

9. Page 5 line 16: A certain amount of 2-bp deletion trp19 mutants 

We have rephrased this sentences as “In T2-generation, the homozygous mutants for null alleles of GmTCP19L frameshift mutated by a 2-bp deletion without any transgenic element resulted in enhanced susceptibility to P. sojae infection by decreasing the activity of antioxidant defense system”.

10. Page 6 line 11: P. sojae race 1 was kindly provided by professor zhang?

We have rephrased this sentences as “P. sojae race 1, PSR01, which is the dominant race in Jilin Province, was kindly provided by Professor Shuzhen Zhang and her team (Soybean Research Institute, Key Laboratory of Soybean Biology of Chinese Education Ministry, Northeast Agricultural University, Harbin, China)”.

11. Page 7 line 1: Having demonstrated that the

We have removed the words “Having demonstrated that”.

12. Page 7 line 6: the Glycine max U6 promoter (GmU6) within its T-DNA region: The Cas9 is not expressed by the U6 promoter.

We have rephrased this sentences as “To construct the GmTCP19L-CRISPR/Cas9 vector carrying both GmTCP19L targeted sgRNA and Cas9 cassettes, the sequence of Cas9 was assembled downstream of the CaMV 35S promoter together with the sgRNA driven by the Glycine max U6 promoter (GmU6) within its T-DNA region”.

13. Page 7 line 7: The bar/basta gene: bar is the Phosphinothricin N-acetyltransferase gene of Streptomyces hygroscopicus. Phosphinothricin is the active component of commercial herbicide Basta.

We have rephrased this sentences as “The bar gene driven by a CaMV 35S promoter was used as a screening marker”.

14. Page 7 line 10: The GmTRP19-CRISPR/Cas9 positive plasmids

We have removed the word “positive”.

15. Page 8 line 5: Simultaneously, we sought to: Transgene-free mutants can only be produced in T1 or T2 progenies through genetic segregation.

I'm sorry that there is not appropriate in our description here. We have rephrased this sentences as “At the same time, we also screened the transgene-free mutants in both T1 and T2 progenies”.

16. Page 8 line 10: potential off-target sites effecting on phenotype,

We have rephrased this sentences as “To examine the specificity of CRISPR/Cas9 in soybean and avoid affecting the phenotype, we analysed the potential off-target sites using online website tool CRISPR-P”.

17. Page 8 line 12: off-target sites for each guide would be selected,

We have rephrased this sentences as “Two most potential off-target sites at GmTCP19L-SP1 and GmTCP19L-SP2 were selected”.

18. Page 8 lines 15-16: to determine whether there were presence of the off-target sites in the amplicon by sequenced.

We have rephrased this sentences as “The regions spanning the target sites were amplified by PCR technique, then the different types of potential off-target sites editing can be identified by sequencing analysis”.

19. Page 8 line 18: The “transgene-clean” trp19 mutant lines: Transgene-free

We have replaced “transgene-clean” trp19 mutant lines” with “Transgene-free mutants”.

20. Page 9 line 6: the significance differences

We have removed the words “the significance differences” and rephrased this sentences as “Non-transformed seedlings were used as controls”.

21. Page 10 line 2: targeted PAM in the exon of GmTRP19 (Fig 1). The CRISPR/Cas9 construct were

We have rephrased this sentences as “Two target sites in the exon of GmTCP19L (named GmTCP19L-SP1 and GmTCP19L-SP2) were chosen (Fig 1), and the corresponding sgRNA/Cas9 vectors were transferred into the soybean cultivar “Williams 82” by Agrobacterium-mediated genetic transformation”.

22. Page 10 lines 12-13: homozygous for null alleles of GmTRP19 and two types of mutations at target site GmTRP19-SP1 (2-bp deletion and 1-bp deletion) were detected (Fig 2A, B). The description is incomplete and not clear. There were homozygous mutant plants detected in three trp19-SP1 T1 lines and also in three trp19-SP2 T1 lines according to Table 1. A total of 14 T1 lines had heterozygous mutant plants. What happened in them? Indels in the target sequences of these homozygous mutant plants should be listed. Indels in the target sequences of the heterozygous mutant plants should also be experimentally determined and then listed.

Thanks very much for your valuable comments. According to Table 1, there were homozygous mutant plants detected in three tcp19l-SP1 T1 lines and also in three tcp19l-SP2 T1 lines. In three tcp19l-SP1 T1 lines, a total of 29 homozygous mutant plants detected by sequencing analysis, only two types of mutations (2-bp deletion and 1-bp deletion, as shown in Figure 2A, B) were obtained. In three tcp19l-SP2 T1 lines, a total of 15 homozygous mutant plants detected by sequencing analysis, only one type of mutations (14-bp deletion, as shown in Figure 2A, C) was obtained. In addition, the heterozygous mutant plants were also determined by sequencing analysis, and the heterozygous mutant plants showed overlapping peaks at the target site (similar to Figure S4), these data have been listed in source data and uploaded.

23. Page 12 lines 18-19: and the part of Cas9 coding sequence in the T-DNA of CRISPR/Cas9 vector was also spanned.

We have rephrased this sentences as “and the Cas9 coding sequence were amplified by PCR technique”.

24. Page 13 line 8: type soybean plant. Labels 1–14, individual mutant lines. Should “lines” be plants?

We have replaced “individual mutant lines” with “individual mutant plants”.

25. Page 13 line 10: T-DNA regions of CRISPR/Cas9 vector.

We have replaced “T-DNA regions of CRISPR/Cas9 vector” with “the Cas9 of sgRNA/Cas9 vectors”.

26. Page 13 line 17: and all of their

We have replaced “their” with “them”.

27. Page 15 line 2: is an important food and economic crops

We have rephrased this sentences as “Soybean [Glycine max (L.) Merr.] is an important food crop with great economic value that abundant protein and oil”.

28. Page 15 line 17: shortened internodes use of the CRISPR/Cas9 system in soybean

We have rephrased this sentences as “Subsequently, the CRISPR/Cas9 system was also used to target four GmLHY genes in soybean, and the height and internodes of the GmLHY mutants were significantly shorter than that of the WT”.

29. Page 15 line 18: obtained E1 gene mutants using CRISPR/Cas9 and soybean transformation,

We have rephrased this sentences as “obtained E1 gene mutants using the CRISPR/Cas9 system and Agrobacterium-mediated transformation technique”.

30. Page 16 lines 1-2: GmTRP19 by CRISPR/Cas9 genome editing approach without any transgenic element of the CRISPR/Cas9 vector, which quickly became a powerful resource for soybean

We have rephrased this sentences as “we screened and obtained homozygous transgene-free mutants targeting soybean endogenous gene GmTCP19L using the CRISPR/Cas9 system, which will significantly increase breeding efficiency and speed up breeding process”.

31. Page 16 line 5: Having demonstrated that

We have removed the words “Having demonstrated that”.

32. Page 16 line 8: which suggesting directions for the breeding or genetic engineering of

We have removed the word “which”.

33. Page 30 line 4: S2 Fig. The basic architecture of the constructs

We have replaced “basic architecture” with “diagram”.

34. S5 Fig. The trp19-SP1-T1#02.03 (2-bp deletion) and trp19-SP1-T1#02.08 (2-bp deletion) are the same sequence since the two plants are derived from the same T1 line. No need to list the same sequence twice. Where is the (1-bp deletion) line or the 14-bp deletion line?

We have checked and corrected the sequence for S6 Fig (S5 Fig in the previous manuscript) in the revised manuscript. In addition, two plants (tcp19l-SP1-T1#02.03 and tcp19l-SP1-T1#02.08) have been proven to be transgene-free mutants, then the sequences of them were listed in S6 Fig. Neither of 1-bp deletion or 14-bp deletion style has been detected in transgene-free mutants, so we did not list the sequences of them in S6 Fig. The full amino acid sequences of them can be seen in in S5 Fig, and the full nucleotide sequences of them can be seen in source data (“Figure 2C-GmTCP19L-SP1(1-bp deletion).ab1” and “Figure 2C-GmTCP19L-SP2(14-bp deletion).ab1”).

35. S1 Table. GmActin primers?

We have added the primers information of the GmActin.

Response to Reviewer#3:

Q1: First, there is no data presented to support the claim that there were no off-target effects from their CRISPR/Cas9 activity. The authors claim there are no off-target effects, yet only provide a supplementary table listing “number of plants with mutations” in off target sites, filled with zeros. The authors describe using the CRISPR-P analysis program, which provides scores based on strength of sequence alignment with a CRISPR construct. The authors ought to describe what threshold they used to identify putative “off-target” candidates, and they also ought to provide some chromatograms to show that these putative off-target sites were not mutated by the CRISPR constructs.

We have provided some chromatograms in Figure 3 to show that the putative off-target sites were not mutated by the CRISPR constructs in the revised manuscript. 

Q2: Second, I don’t think the authors explore enough phenotypes of their mutants. The authors acknowledge at many points in the manuscript that this class of transcription factor play a wide range of roles in growth, development, flowering, circadian rhythm, etc. But, the authors only present one phenotype – susceptibility to 1 isolate of P. sojae. Therefore, the increased susceptibility to P. sojae could be one of many phenotypes of these mutant lines, which could compromise the conclusion that GmTRP19 has a direct role in resistance to P. sojae. Additionally, the phenotype that the authors do test are only observational, with no numeric data or statistics to suppor the conclusion of increased susceptibility in the mutants.

We are very grateful to you for your valuable suggestions. In previous studies, there were reports that a few of TCP genes also play a wide range of roles in leaf development (Efroni et al, 2008), leaf morphogenesis (Palatnik et al., 2003), leaf senescence (Schommer et al., 2008), trichome development (Steiner et al., 2012; Vadde et al., 2018) flower development (Nag et al., 2009; Koyama et al., 2011; Li et al., 2021) and circadian clock (Giraud et al., 2010). So, in the follow-up work, it is valuable that exploring enough phenotypes of the mutants, and it will improve the interpretation of the findings that the function of the GmTCP19L response to P. sojae infection. In the revised manuscript, we supplemented this relevant content in the discussion section.

Q3: Specific comments:

1. Page 2, Line 15: Remove “the” before “conserved domains”

We have removed the word “the”.

2. Page 4, Line 3: What is the FT-FD complex. Please list in full before using an abbreviation. Page 4, Line 4: What is the AP1 gene, and what is the relevance of AtTCP12 and AtTCP18 binding to it?

We have rephrased this sentences as “AtTCP12/AtTCP18 can integrate into the FLOWERING LOCUS T (FT)‒FD complex to control floral initiation and also directly bind the promoter of downstream floral meristem identity gene APETALA1(AP1) to enhance its transcription and regulate flowering”.

3. Page 4, Line 17: Hyaloperonospora arabidopsidis is not a gram-negative bacteria. It is an oomycete. Please correct.

We have corrected it.

4. Page 5, Line 9: Remove “related to P. sojae”. You describe the connection between the gene and P. sojae later in the sentence.

We have removed the words “related to P. sojae”.

5. Page 5, Line 18: Remove “the” before “resistance”. There is no singular resistance to P. sojae – there is single, R-gene mediated resistance, and partial resistance (Dorrance, 2018 doi.org/10.1080/07060661.2018.1445127). The GmTPS19 gene is not an R-gene.

We have removed the word “the” before “resistance”.

6. Page 6, line 6: Williams 82 has high partial resistance to P. sojae, but only 1 R-gene mediating robust resistance to specific pathotypes of P. sojae (Dorrance, 2018).

In previous studies, there were reports that a few of genes were induced by P. sojae, such as N-rich protein (Ludwig and Tenhaken, 2001), GmPR10 (Xu et al., 2014), GmPIB1 (Dong et al., 2018) and GmBTB (Zhang et al., 2021). Thus, we speculate that there maybe P. sojae induced genes changed in their expression when the expression of GmTCP19L is modified, which might affect the resistance of soybean response to P. sojae infection. So, in the follow-up work, it is valuable that making a deeper research on protein interaction and regulatory network building. In the revised manuscript, we supplemented this relevant content in the discussion section. 

7. Page 6, line 11: Capitalize “Professor Zhang”.

We have replaced “professor” with “Professor”.

8. Page 8, line 6: What is a PAT/Bar “test strip”? Did you spray these plants with gufosinate? Did you do PCR to detect the presence / absence of this gene? Did you get strips from a company? If so, where? More detail needed here for reproducibility.

We have rephrased this sentences as “PAT/Bar test strip (Catalog No. STX 14200, Agdia, USA) was used to identify the selective marker bar gene”.

9. Page 8, line 12: “Two off-target sites for each guide would be selected” does this mean there were 2 off target sites detected by CRISPR-P analysis? Or only 2 were selected for analysis of off-target effects? There’s a big difference, please clarify.

Page 12, lines 8-9: How “likely” were these two off-target sites predicted to be affected? How many other off-target sites were identified at a lower “likelihood”? Can you provide the results of the CRISPR-P analysis in the supplemental files, or list here in some form?

I'm sorry that there is not appropriate in our description here. And the most potential off-target analysis at the two target sites of GmTCP19L in the T1 generation have been listed in S2 Table. Meantime, we have provided some chromatograms in Figure 3 to show that the putative off-target sites were not mutated by the CRISPR constructs in the revised manuscript. 

10. Page 10, lines 7-8: Please clarify the language here a bit. You list two numbers and use “respectively”, but don’t list both conditions. Please change to something like this “…had heterozygous mutations at the first (SP1) and second (SP2) target sites, respectively (S3 Fig in S1 File)”.

We have checked and corrected this problem.

11. Page 10, line 16: Please don’t use “PTCs” as an abbreviation. This is unnecessary.

We have removed the abbreviation “PTCs”.

12. Page 12, lines 12-13: “no mutations were observed” yet the data to support this is just a table with 0’s listed. Please provide the chromatograms to show evidence that there were no off-target effects on DNA sequence.

We have provided some chromatograms in Figure 3 to show that the putative off-target sites were not mutated by the CRISPR constructs in the revised manuscript.

13. Page 13, Line 17: “all of their…” all of their what? I think there is a word missing here. Or perhaps “their” is meant to be “them”?

We have replaced “their” with “them”.

14. Figure 4: These soybeans do not look like they are growing in vermiculite, as described throughout the manuscript. Explain.

In low-generation, the number of homozygous plants obtained by screening is small, and one of the most important purposes of this research is to obtain new germplasm materials for breeding research, in this case, in order to improve survival rate of the mutant plants, seeds collected from the T0 generation were sown in pots filled with sterile vermiculite. In the T2 generation, for disease resistance analysis, seeds collected from T1 ‘transgene-free’ mutants were sown in pots filled with nutritious soil and grown in a greenhouse. So, there is an error in our description here, and we have checked and corrected the description in the revised manuscript.

15. Page 15, lines 15-19: This brings up a great point. Did any of your GmTRP19 mutants display any other phenotypes besides increased sensitivity to P. sojae? Were there any developmental phenotypes (delayed germinating, effects on plant height, flowering, etc.)?

We are very grateful to you for your valuable suggestions. In previous studies, there were reports that a few of TCP genes also play a wide range of roles in leaf development (Efroni et al, 2008), leaf morphogenesis (Palatnik et al., 2003), leaf senescence (Schommer et al., 2008), trichome development (Steiner et al., 2012; Vadde et al., 2018) flower development (Nag et al., 2009; Koyama et al., 2011; Li et al., 2021) and circadian clock (Giraud et al., 2010). So, in the follow-up work, it is valuable that exploring enough phenotypes of the mutants, and it will improve the interpretation of the findings that the function of the GmTCP19L response to P. sojae infection. In the revised manuscript, we supplemented this relevant content in the discussion section.

16. Page 16, line 4: What makes GmTRP19 a “new” member of this family?

We have rephrased this sentences as “GmTCP19L is the first TCP gene was identified in soybean response to P. sojae infection:.

17. Page 16, lines 5-9: This entire sentence is an incomplete sentence. Authors could remove “Having demonstrated that”, to make it complete.

We have rephrased this sentences as “In previous studies, TCP transcription factors have been shown to be play a versatile regulatory role at the different stage of plant growth and development”.

18. “We obtained a novel type of mutations”

We have rephrased this sentences as “We obtained the transgene-free mutants with 2-bp deletion at GmTCP19L coding region, and the frameshift mutations produced premature translation termination codons and truncated GmTCP19L proteins, increasing susceptibility to P. sojae infection in the T2-generation”.

19. “which suggesting”

We have removed the word “which”.

20. “this gene resistance to P. sojae”

We have rephrased this sentences as “studied the function of this gene responses to P. sojae infection”.

21. “exhibited more sensitive to P. sojae”

We have rephrased this sentences as “In T2-generation, the homozygous mutants for null alleles of GmTCP19L frameshift mutated by a 2-bp deletion without any transgenic element resulted in enhanced susceptibility to P. sojae infection by decreasing the activity of antioxidant defense system”.

22. “All of three types”

We have rephrased this sentences as “The frameshift mutations of three types at two target sites of GmTCP19L”.

23. “an important food and economic crops”

We have rephrased this sentences as “Soybean [Glycine max (L.) Merr.] is an important food crop with great economic value that abundant protein and oil”.

---

## [Decision Letter · Decision Letter 1]

23 Feb 2022

PONE-D-21-31763R1 CRISPR/Cas9-mediated targeted mutagenesis of GmTCP19L increasing susceptibility to Phytophthora sojae in soybean

Dear Dr. Fan,

Thank you for re-submitting your manuscript to PLOS ONE. After careful consideration, we feel that it has merit for publication and we invite you to submit a revised version of the manuscript that addresses the points raised during the review process.

We look forward to receiving your revised manuscript.

Kind regards,

Hao-Xun Chang, Ph.D.

Academic Editor

PLOS ONE

Journal Requirements:

**Additional Editor Comments:**

The manuscript has been greatly improved in clarity, but some descriptions and methods still need minor revisions and additional details.

Page 2 line 1. Delete (P. sojae)

Page 2 line 4. The Abstract should be revised. If the authors already known that GmTCP19L controlling soybean resistance to P. sojae, there will be no need for the rest of study presented in the manuscript. The sentence here should be the background, not the conclusion. Description such as “and generated targeted mutants of GmTCP19L gene, which was previously related to involve in soybean responses to P. sojae” would make more sense.

Page 2 line 10. The authors emphasized multiple times throughout the manuscript regarding the mutants without "any transgenic elements." This is a bold statement as the authors only detect BAR and a PCR fragment of Cas9 vector. In other words, if there is any other part of vector elements remained in the mutant genome, the methods will have no capability of detecting them. Therefore, I suggest the authors to revise these statements in a conserved way, e.g. the BAR and Cas9 sequences were undetectable using strip and PCR methods, respectively. Others: Page 6 line 1, Page 9 line 6, Page 10 line 12, Page 14 line 15 (Figure 4 title), Page 15 line 6, Page 20 line 12 etc.

Page 2 line 14. TCP should be spell-out for clarity especially in the every beginning of the text.

Page 3 line 2. Is “shown” to

Page 3 line 4. classified “into” Class I

Page 3 line 9. Italize cis-

Page 3 line 15. Have been found, “and” most of which

Page 3 line 18 to Page 4 line 3. This is a long sentence with many information. Please simplify or separate it.

Page 4 line 14. No need for (BR) because it is not mentioned later on.

Page 4 line 19 to Page 5 line 1. Please double confirm the meaning of this sentence. It reads that two pathogens target TCP to enhance disease resistance…Would the pathogens do these functions against themselves?

Page 5 line 10. The authors used the W82a1v1 names throughout the manuscript. However, the popular and common names of soybean genes are already based on the W82a2v1 version, or even a4v1 version. Please revise the name or at least provide W82a2v1 names side-by-side. Others: Page 7 line 2, Figure 3 etc. Please also confirm the name used in the phylogentic tree, it seems to be the a2v1 name.

Page 5 line 14. Arabidopsis, “and” this

Page 5 line 16. In “the” soybean cultivar

Page 5 line 18. “with” short deletions

Page 6 line 4. soybean resistance to P. sojae.

Page 6 line 10. “high resistance to P. sojae” is not a specific or scientific description. How high is high? Please be specific, e.g. W82 carries Rps-1k? or what do the authors mean for high?

Page 7 line 9-12. Please provide the vector plot with clear locations of primers, the BAR resistance gene, the cloning regions for GmTCP19L, and the detection region for Cas9.

Page 7 line 15. More details needed for tri-parental mating.

Page 8 line 8-9. Delete the parenthesis. Frameshift mutation is clear enough.

Page 8 line 10-13. Please specify the length or region of BAR detection and Cas9 amplicon.

Page 9 line 17-18. More details of the methods are needed. How as the P. sojae quantified? Disease severity? Counting sporangia? qPCR? In addition, what was the statistical method?

Page 10 line 4. One gram

Page 10 line 7. Which part of results were analyzed using “multiple pairwise comparison”? In addition, was the assumption of t test checked? Please specify.

Page 11 line 20. Replace “. Simultaneously” to “; meanwhile,”

Page 14 line 1. Please provide figure legend for Figure 3. Please verify the version of soybean gene name.

Page 15 line 5. In response to

Page 15 line 6-7. Are these two T1 lines, tcp19l-SP1-T1 7 (2-bp deletion), tcp19l-SP1-T2), identical to “tcp19l-SP1-T1#02.03 and tcp19l-SP1-T1#02.08”? Or which is which? Please clarify.

Page 16 line -2. Methods for qualification needed.

Page 16 line 9. Please provide figure legend for Figure 5.

Page 16 line 15 to Page 17 line 3. If this is the quantification method, please move to M&M, and specify the primers and provide methods e.g. construction of standard curve? for details.

Page 18 line 16 to Page 19 line 1.

Page 18 line 12. RNA-Seq

Page 19 line 20 to Page 20 line 1. Please provide previse literature for supporting ROS involves in P. sojae resistance.

Page 20 line 11. Delete Now.

In Figure 5 and 6, please clarify which lineage does the tcp19l-SP1-T2 and tcp19l-SP1-T3 from? Are they 2-bp or 1-bp deletion mutants? Are they progenies of tcp19l-SP1-T1#02.03 and tcp19l-SP1-T1#02.08? Have different mutant lines/lineage being tested for resistance? Please note if only one mutant lineage was used in pathogenicity test, the results may not be convincing. If different mutant lineage, e.g. 2-bp and 1-bp, were tested and the results were consistent, the conclusion will be more reliable.

In addition to these editorial suggestions, the reviewers would like to know more about the phenotypes of 2-bp or 1-bp deletion mutants, and/or tcp19l-SP1-T1#02.03 and tcp19l-SP1-T1#02.08. The reviewers’ opinion is highly respected because if there are other phenotypic defects, the damping-off may be a side-effect due to other genetic problems instead of purely disease susceptibility. Your feedback in the discussion included several literatures mostly based on Arabidopsis (Page 18 line 16-20), but not direct observation on your T0, T1, T2, or T3 soybean mutants. Therefore, I would like to invite you to provide more details for the final editorial and content improvement.

**Reviewers' comments:**

The authors have addressed all but 1 concern from previous review. As I noted, and as noted by the other reviewer, and acknowledged by the authors themselves, mutating a transcription factor can have broad effects. Yet, the only phenotype examined was "resistance to P. sojae". While the authors acknowledge that they intend to look for other phenotypes in follow up work, it would be beneficial here.

Only a few small remaining comments that, if addressed, make it suitable for publication. The grammar is improved, though some issues are still present.

Page 11, lines 7-8: Here, GmTCP19L-SP1 and GmTCP19L-SP2 are described as “target sites”. Yet an entire section on page 8 lines 14-20 is dedicated to the description of “off-target sites” called the same thing. Please clarify what is meant by “off-target sites” throughout the manuscript. I believe you designed GmTCP19L-SP1 and GmTCP19L-SP2 as “target sites”, and then looked to see if either of them could have off-target activity.

Remove the red dashed circles from Figure 5.

Provide units / description for Y-axis in Figure 6B. What does “Accumulation” represent? Number of zoospores re-isolated? Quantity of P. sojae DNA per µg of plant DNA?

---

## [Author Response · Author response to Decision Letter 1]

8 Apr 2022

Response to Editor：

1. Page 2 line 1. Delete (P. sojae)

We have removed the abbreviation “P. sojae”.

2. Page 2 line 4. The Abstract should be revised. If the authors already known that GmTCP19L controlling soybean resistance to P. sojae, there will be no need for the rest of study presented in the manuscript. The sentence here should be the background, not the conclusion. Description such as “and generated targeted mutants of GmTCP19L gene, which was previously related to involve in soybean responses to P. sojae” would make more sense.

We have rephrased this sentences as “Agrobacterium-mediated transformation was used to introduce the CRISPR/Cas9 expression vector into soybean cultivar ‘Williams 82’ and generated targeted mutants of GmTCP19L gene, which was previously related to involve in soybean responses to P. sojae”.

3. Page 2 line 10. The authors emphasized multiple times throughout the manuscript regarding the mutants without "any transgenic elements." This is a bold statement as the authors only detect BAR and a PCR fragment of Cas9 vector. In other words, if there is any other part of vector elements remained in the mutant genome, the methods will have no capability of detecting them. Therefore, I suggest the authors to revise these statements in a conserved way, e.g. the BAR and Cas9 sequences were undetectable using strip and PCR methods, respectively. Others: Page 6 line 1, Page 9 line 6, Page 10 line 12, Page 14 line 15 (Figure 4 title), Page 15 line 6, Page 20 line 12 etc.

We appreciate very much the valuable suggestions from you, and we have rephrased this sentences as “The new soybean germplasm with homozygous tcp19l mutations but the BAR and Cas9 sequences were undetectable using strip and PCR methods, respectively” or “mutants without the BAR and Cas9 sequences of the CRISPR/Cas9 vector”.

4. Page 2 line 14. TCP should be spell-out for clarity especially in the every beginning of the text.

TCP has been spelled out clearly in the beginning of the text as you suggested.

5. Page 3 line 2. Is “shown” to

We have replaced “thought” with “shown”.

6. Page 3 line 4. classified “into” Class I

We have added the word “into” as you suggested.

7. Page 3 line 9. Italize cis-

We have made the correction based on your suggestion.

8. Page 3 line 15. Have been found, “and” most of which

We have added the word “and” as you suggested.

9. Page 3 line 18 to Page 4 line 3. This is a long sentence with many information. Please simplify or separate it.

We have rephrased this sentences as “AtTCP14 and AtTCP15 can modulate cell proliferation in the developing leaf blade and specific floral tissues[26], and also participate in the induction of genes involved in gibberellin biosynthesis and cell expansion by high temperature functionally [27, 28]”. 

10. Page 4 line 14. No need for (BR) because it is not mentioned later on.

We have removed the abbreviation “(BR)”.

11. Page 4 line 19 to Page 5 line 1. Please double confirm the meaning of this sentence. It reads that two pathogens target TCP to enhance disease resistance…Would the pathogens do these functions against themselves?

We have checked and corrected the sentences as “AtTCP13, AtTCP15 and AtTCP19 can be targeted by effectors from the gram-negative bacteria Pseudomonas syringae and the oomycete Hyaloperonospora arabidopsidis, and the plants mutated in AtTCP13, AtTCP15 or AtTCP19 exhibit altered infection phenotypes”.

12. Page 5 line 10. The authors used the W82a1v1 names throughout the manuscript. However, the popular and common names of soybean genes are already based on the W82a2v1 version, or even a4v1 version. Please revise the name or at least provide W82a2v1 names side-by-side. Others: Page 7 line 2, Figure 3 etc. Please also confirm the name used in the phylogentic tree, it seems to be the a2v1 name.

We have checked and corrected the name as Glyma.05G050400.1 based on the W82a2v1 version.

13. Page 5 line 14. Arabidopsis, “and” this

We have replaced “And then” with “and”.

14. Page 5 line 16. In “the” soybean cultivar

We have added the word “the”.

15. Page 5 line 18. “with” short deletions

We have replaced “by” with “with”.

16. Page 6 line 4. soybean resistance to P. sojae.

We have rephrased this sentences as “GmTCP19L directly or indirectly regulates soybean resistance to P. sojae”.

17. Page 6 line 10. “high resistance to P. sojae” is not a specific or scientific description. How high is high? Please be specific, e.g. W82 carries Rps-1k? or what do the authors mean for high?

We have rephrased this sentences as “Williams 82, a P. sojae-resistant soybean cultivar carrying resistance gene Rps1k, was used for transformation in this study”.

18. Page 7 line 9-12. Please provide the vector plot with clear locations of primers, the BAR resistance gene, the cloning regions for GmTCP19L, and the detection region for Cas9.

The vector plot with clear locations of target sites of GmTCP19L, the primers of the detection region for Cas9, and the BAR resistance gene was shown in S2 Fig.

19. Page 7 line 15. More details needed for tri-parental mating.

Suggested by Reviewer, the method of plasmid delivery by tri‐parental mating were specified in the M&M section of the revised manuscript. 

20. Page 8 line 8-9. Delete. Frameshift mutation is clear enough.

We have removed the parenthesis “(not multiples of three)”. 

21. Page 8 line 10-13. Please specify the length or region of BAR detection and Cas9 amplicon.

The BAR detection and the region of Cas9 amplicon were specified in the M&M section, we have rephrased this sentences as “PAT/Bar test strip (Catalog No. STX 14200, Agdia, USA) was used to identify the BAR protein following the manufacturer’s instruction, and the primers Cas9-F/R were used to amplify the fragment (349 bp) of the Cas9 (S2 Fig in S1 File)”..

22. Page 9 line 17-18. More details of the methods are needed. How as the P. sojae quantified? Disease severity? Counting sporangia? qPCR? In addition, what was the statistical method?

Thanks very much for your valuable comments. There was not a detailed description for pathogen level analysis, and we added the detailed description in the revised manuscript. The relative accumulation of P.sojae in infected cotyledons and roots were also analyzed by qPCR using One Step RT-PCR Kit (Code No.:PCR-311, TOYOBO, Japan) on a QuantStudio 3 instrument (Thermo, United States). The expression of the P. sojae housekeeping gene PsACT (GenBank accession no. XM_009530461.1) relative to the soybean housekeeping gene GmCYP2 (Glyma.12G024700) (ΔCt = CtHK of P. sojae - CtHK of soybean) was calculated. The housekeeping genes of P. sojae and soybean were chosen as described previously by Wang et al. (2011) , and the primers were listed in S1 Table (in S1 File).

23. Page 10 line 4. One gram

We have replaced “one gram” with “1 g”.

24. Page 10 line 7. Which part of results were analyzed using “multiple pairwise comparison”? In addition, was the assumption of t test checked? Please specify.

I'm sorry that there is an error in our description here, and we have checked and corrected the description in the revised manuscript. Relative enzyme activity, i.e. the ratio of SOD activity or POD activity in infected roots under P. sojae zoospores in tcp19l mutants versus non-transformed seedlings at the same time point were measured. Three biological replicates, each with three technical replicates, were averaged and statistically analysed using Student's t-test (*P<0.05, **P<0.01). Bars indicate the standard deviation of the mean.

25. Page 11 line 20. Replace “. Simultaneously” to “; meanwhile,”

We have replaced “Simultaneously” to “Meanwhile”.

26. Page 14 line 1. Please provide figure legend for Figure 3. Please verify the version of soybean gene name.

We have supplemented the figure legend in Figure 3 and corrected the soybean gene name based on the W82a2v1 version.

27. Page 15 line 5. In response to

We have rephrased this sentences as “Phenotypes of the mutants in response to P. sojae infection”. We have replaced “response to” to “in response to”.

28. Page 15 line 6-7. Are these two T1 lines, tcp19l-SP1-T1 7 (2-bp deletion), tcp19l-SP1-T2), identical to “tcp19l-SP1-T1#02.03 and tcp19l-SP1-T1#02.08”? Or which is which? Please clarify.

We have rephrased this sentences as “In the T2 generation, the progeny of T1 homozygous tcp19l mutants tcp19l-SP1-T1#02.03 and tcp19l-SP1-T1#02.08 were grown under natural conditions. Because of the mutant of them were consistent, we number them uniformly as tcp19l-SP1-T2, and all of them were homozygous tcp19l mutants without the BAR and Cas9 sequences (Table 2)”.

29. Page 16 line -2. Methods for qualification needed.

There was not a detailed description for pathogen level analysis, and we added the detailed description in the M&M section of the revised manuscript. The relative accumulation of P.sojae in infected cotyledons and roots were also analyzed by qPCR using One Step RT-PCR Kit (Code No.:PCR-311, TOYOBO, Japan) on a QuantStudio 3 instrument (Thermo, United States). The expression of the P. sojae housekeeping gene PsACT (GenBank accession no. XM_009530461.1) relative to the soybean housekeeping gene GmCYP2 (Glyma.12G024700) (ΔCt = CtHK of P. sojae - CtHK of soybean) was calculated. The housekeeping genes of P. sojae and soybean were chosen as described previously by Wang et al. (2011) , and the primers were listed in S1 Table (in S1 File).

30. Page 16 line 9. Please provide figure legend for Figure 5.

We have supplemented the figure legend in Figure 5.

31. Page 16 line 15 to Page 17 line 3. If this is the quantification method, please move to M&M, and specify the primers and provide methods e.g. construction of standard curve? for details.

We have provided the detailed description for pathogen level analysis in the M&M section of the revised manuscript. The relative accumulation of P.sojae in infected cotyledons and roots were also analyzed by qPCR using One Step RT-PCR Kit (Code No.:PCR-311, TOYOBO, Japan) on a QuantStudio 3 instrument (Thermo, United States). The expression of the P. sojae housekeeping gene PsACT (GenBank accession no. XM_009530461.1) relative to the soybean housekeeping gene GmCYP2 (Glyma.12G024700) (ΔCt = CtHK of P. sojae - CtHK of soybean) was calculated. The housekeeping genes of P. sojae and soybean were chosen as described previously by Wang et al. (2011), and the primers were listed in S1 Table (in S1 File).

32. Page 18 line 12. RNA-Seq

We have replaced “RNA-seq” with “RNA-Seq”

33. Page 19 line 20 to Page 20 line 1. Please provide previse literature for supporting ROS involves in P. sojae resistance.

GmPIB1, a bHLH-type transcription factor, has been shown to facilitate resistance to P. sojae in Glycine max by affecting ROS levels. So, we provided this previse literature for supporting ROS involves in P. sojae resistance in the revised manuscript.

34. Page 20 line 11. Delete Now.

We have removed the word “Now”.

35. In Figure 5 and 6, please clarify which lineage does the tcp19l-SP1-T2 and tcp19l-SP1-T3 from? Are they 2-bp or 1-bp deletion mutants? Are they progenies of tcp19l-SP1-T1#02.03 and tcp19l-SP1-T1#02.08? Have different mutant lines/lineage being tested for resistance? Please note if only one mutant lineage was used in pathogenicity test, the results may not be convincing. If different mutant lineage, e.g. 2-bp and 1-bp, were tested and the results were consistent, the conclusion will be more reliable.

Thanks very much for your valuable suggestions. And I’m sorry to tell you that because of the mutant of T1 tcp19l mutants tcp19l-SP1-T1#02.03 and tcp19l-SP1-T1#02.08 were consistent (2-bp deletion), we number the progeny of them uniformly as tcp19l-SP1-T2, and tcp19l-SP1-T3 from tcp19l-SP1-T2. We missed resistance identification for 1-bp deletion mutants in the T2 generation because of we eliminated them when screening “transgene-free” mutants. But, in the T1 generation, three types of tcp19l mutants (1-bp deletion and 2-bp deletion at target site GmTCP19L-SP1, 14-bp deletion at target site GmTCP19L-SP2) were detected by sequence alignment with the WT sequence. At that time, we considered that the number of tcp19l mutants is small, and one of the most important purposes of this research is to obtain new germplasm materials for breeding research, so, in order to avoid death of tcp19l mutants after P. sojae infection, the living cotyledons of three types of tcp19l mutants were selected to investigate resistance to P. sojae. After 3 days of incubation with zoospores of P. sojae, a remarkable difference in the development of disease symptoms was observed. In tcp19l mutants, the cotyledons became soft and exhibited clear water-soaked lesions compared with those of the WT, and the relative biomass of P. sojae in infected living cotyledons was significantly (P<0.05) higher in tcp19l mutants (1-bp deletion and 2-bp deletion) than that in WT. Because of we did not report these results in previous manuscript, we supplemented this relevant content in the revised manuscript.

36. In addition to these editorial suggestions, the reviewers would like to know more about the phenotypes of 2-bp or 1-bp deletion mutants, and/or tcp19l-SP1-T1#02.03 and tcp19l-SP1-T1#02.08. The reviewers’ opinion is highly respected because if there are other phenotypic defects, the damping-off may be a side-effect due to other genetic problems instead of purely disease susceptibility. Your feedback in the discussion included several literatures mostly based on Arabidopsis (Page 18 line 16-20), but not direct observation on your T0, T1, T2, or T3 soybean mutants. Therefore, I would like to invite you to provide more details for the final editorial and content improvement.

We are very grateful to you for your valuable suggestions. In the low generation, the number of tcp19l mutants is small, and one of the most important purposes of this research is to obtain new germplasm materials for breeding research, so, we missed more phenotypes identification for mutants. But, we identified the off-target sites in the progeny of tcp19l mutants and no mutations were observed in the examined potential off-target sites, meanwhile, we did not find any other phenotypic changes from T0, T1, T2, or T3 tcp19l mutants. In the follow-up work, we will continue to multiply the number of seeds and make a deeper research on the phenotypes of tcp19l mutants, such as leaf morphogenesis, leaf senescence, flower development, hormone signaling, and response to other stresses. In the discussion section, we supplemented this relevant content in the revised manuscript.

Response to Reviewers：

1. Page 11, lines 7-8: Here, GmTCP19L-SP1 and GmTCP19L-SP2 are described as “target sites”. Yet an entire section on page 8 lines 14-20 is dedicated to the description of “off-target sites” called the same thing. Please clarify what is meant by “off-target sites” throughout the manuscript. I believe you designed GmTCP19L-SP1 and GmTCP19L-SP2 as “target sites”, and then looked to see if either of them could have off-target activity.

Like you said, we designed GmTCP19L-SP1 and GmTCP19L-SP2 as “target sites”, and then looked to see if either of target sites could have off-target activity, we analysed the potential off-target sites using online website tool CRISPR-P. We have supplemented the relevant description in the M&M section of the revised manuscript as you suggested.

2. Remove the red dashed circles from Figure 5.

We have removed the red dashed circles from Figure 5 as you suggested.

3. Provide units / description for Y-axis in Figure 6B. What does “Accumulation” represent? Number of zoospores re-isolated? Quantity of P. sojae DNA per µg of plant DNA?

Here is a relative expression. I'm sorry that I didn't describe the calculation method very well before. We have provided the detailed description for pathogen level analysis in the M&M section of the revised manuscript. The relative accumulation of P.sojae in infected cotyledons and roots were also analyzed by qPCR using One Step RT-PCR Kit (Code No.:PCR-311, TOYOBO, Japan) on a QuantStudio 3 instrument (Thermo, United States). The expression of the P. sojae housekeeping gene PsACT (GenBank accession no. XM_009530461.1) relative to the soybean housekeeping gene GmCYP2 (Glyma.12G024700) (ΔCt = CtHK of P. sojae - CtHK of soybean) was calculated. The housekeeping genes of P. sojae and soybean were chosen as described previously by Wang et al. (2011), and the primers were listed in S1 Table.

---

## [Editor Report · Decision Letter 2]

11 Apr 2022

CRISPR/Cas9-mediated targeted mutagenesis of GmTCP19L increasing susceptibility to Phytophthora sojae in soybean

PONE-D-21-31763R2

Dear Dr. Fan,

We’re pleased to inform you that your manuscript has been judged scientifically suitable for publication and will be formally accepted for publication once it meets all requirements.

Kind regards,

Hao-Xun Chang, Ph.D.

Academic Editor

PLOS ONE

---

## [Editor Report · Acceptance letter]

1 Jun 2022

PONE-D-21-31763R2 

CRISPR/Cas9-mediated targeted mutagenesis of *GmTCP19L* increasing susceptibility to *Phytophthora sojae* in soybean 

Dear Dr. Fan:

I'm pleased to inform you that your manuscript has been deemed suitable for publication in PLOS ONE. Congratulations! Your manuscript is now with our production department. 

Kind regards, 

on behalf of

Dr. Hao-Xun Chang 

Academic Editor

PLOS ONE